# Motoneuron Wnts regulate neuromuscular junction development

Chengyong Shen[1†]*, Lei Li[2†], Kai Zhao[3†], Lei Bai[1†], Ailian Wang[1], Xiaoqiu Shu[1], Yatao Xiao[1], Jianmin Zhang[1], Kejing Zhang[1], Tiankun Hui[4], Wenbing Chen[2,4], Bin Zhang[5], Wei Hsu[6], Wen-Cheng Xiong[2,7], Lin Mei[2,7]*

[1]Department of Neurology, the First Affiliated Hospital, Institute of Translational Medicine, School of Medicine, Zhejiang University, Zhejiang, China; [2]Department of Neurosciences, School of Medicine, Case Western Reserve University, Cleveland, Ohio, United States; [3]Department of Neuroscience and Regenerative Medicine, Medical College of Georgia, Augusta University, Augusta, Georgia, United States; [4]Institute of Life Science, Nanchang University, Nanchang, Jiangxi, China; [5]Department of Physiology, School of Basic Medicine, Institute of Brain Research, Huazhong University of Science and Technology, Wuhan, Hubei, China; [6]Department of Biomedical Genetics, Center for Oral Biology, James Wilmot Cancer Center, University of Rochester Medical Center, Rochester, New York, United States; [7]Louis Stokes Cleveland Veterans Affairs Medical Center, Cleveland, Ohio, United States

*For correspondence:
cshen@zju.edu.cn (CS);
lin.mei@case.edu (LM)

[†]These authors contributed equally to this work

Competing interests: The authors declare that no competing interests exist.

**Abstract** The neuromuscular junction (NMJ) is a synapse between motoneurons and skeletal muscles to control motor behavior. Unlike extensively investigated postsynaptic differentiation, less is known about mechanisms of presynaptic assembly. Genetic evidence of Wnt in mammalian NMJ development was missing due to the existence of multiple Wnts and their receptors. We show when Wnt secretion is abolished from motoneurons by mutating the Wnt ligand secretion mediator (Wls) gene, mutant mice showed muscle weakness and neurotransmission impairment. NMJs were unstable with reduced synaptic junctional folds and fragmented AChR clusters. Nerve terminals were swollen; synaptic vesicles were fewer and mislocated. The presynaptic deficits occurred earlier than postsynaptic deficits. Intriguingly, these phenotypes were not observed when deleting Wls in muscles or Schwann cells. We identified Wnt7A and Wnt7B as major Wnts for nerve terminal development in rescue experiments. These observations demonstrate a necessary role of motoneuron Wnts in NMJ development, in particular presynaptic differentiation.
DOI: https://doi.org/10.7554/eLife.34625.001

## Introduction

The neuromuscular junction (NMJ) is a peripheral synapse between motoneurons and skeletal muscle fibers that is critical to control muscle contractions. Dysfunction of this synapse has been implicated in various neurological disorders including congenital myasthenia syndrome (CMS), myasthenia gravis (MG), and amyotrophic lateral sclerosis (ALS) (*Li et al., 2018*; *Sanes and Lichtman, 2001*; *Shen et al., 2015*; *Wu et al., 2010*). During NMJ formation, acetylcholine receptors (AChRs) become concentrated at the postjunctional membrane whereas axon terminals differentiate to form active zones that align precisely to AChR clusters on junctional folds. NMJ development requires intimate interaction between motor nerve terminals and muscle fibers. For example, motoneurons produce agrin that binds its receptor LRP4 (low-density lipoprotein receptor-related protein 4) in muscle fibers and activates the muscle specific kinase MuSK (*Bezakova and Ruegg, 2003*;

*DeChiara et al., 1996*; *Kim et al., 2008*; *McMahan, 1990*; *Zhang et al., 2008*).The agrin-LRP4-MuSK signaling is necessary for NMJ formation, in particular postsynaptic AChR clustering. In contrast to postsynaptic differentiation, much less is known about mechanisms of presynaptic assembly of the NMJ.

Recent evidence suggests that Wnt ligands may regulate synapse development in mammals. Regarding NMJ development, Wnts can induce AChR clusters by themselves and can promote agrin-induced AChR clustering in cultured C2C12 myotubes (*Barik et al., 2014b*; *Zhang et al., 2012*). These effects seem to require the extracellular cysteine-rich domain (CRD) of MuSK, which is homologous to that in the Wnt receptor Frizzled (*Hubbard and Gnanasambandan, 2013*). Deleting the CRD was shown to impair in vivo NMJ formation (*Messéant et al., 2015*; *Strochlic et al., 2012*) although this notion was challenged by another study (*Remédio et al., 2016*). Intracellularly, MuSK interacts with disheveled (Dvl1), the key scaffold protein in the Wnt signal pathway (*Luo et al., 2002*). In Zebrafish, Wnt binding to the CRD of MuSK and MuSK interaction with Dvl are necessary for guiding axons to the middle region of muscle fibers (*Jing et al., 2009*). Mutant mice lacking or overexpressing β-catenin, a key mediator of the Wnt canonical pathway, in muscle fibers display both pre- and postsynaptic deficits (*Li et al., 2008*; *Liu et al., 2012*; *Wu et al., 2012a*). In terms of synapses in the brain, Dvl1 expresses higher in axons than dendrites, especially in growth cones of cultured hippocampus neurons, suggesting a role of Wnt pathway in presynaptic differentiation (*Zhang et al., 2007*). Cerebella granule cells secret Wnt7A to induce axon and growth cone remodeling, including Synapsin 1 puncta clustering (*Hall et al., 2000*). Inducing the expression of Dickkopf1, a secreted Wnt antagonist, decreases the number of cortico-striatal synapses in adult mice. The remaining excitatory terminals contain fewer synaptic vesicles (*Galli et al., 2014*).

There are 19 different Wnts in rodents and humans (*Barik et al., 2014b*), many of which have redundant functions, making it difficult to reveal convincingly the function of Wnt ligands in vivo. In this study, we investigated the role of Wnt ligands in NMJ formation by taking the advantage of Wnt ligand secretion mediator (Wls). Wls is a seven-transmembrane protein that is necessary for sorting and transporting Wnts from the endoplasmic reticulum to the cell surface (*Hausmann et al., 2007*). In Drosophila Wls mutants, Wingless (the homolog of vertebrate Wnt1) is retained intracellularly and cannot be

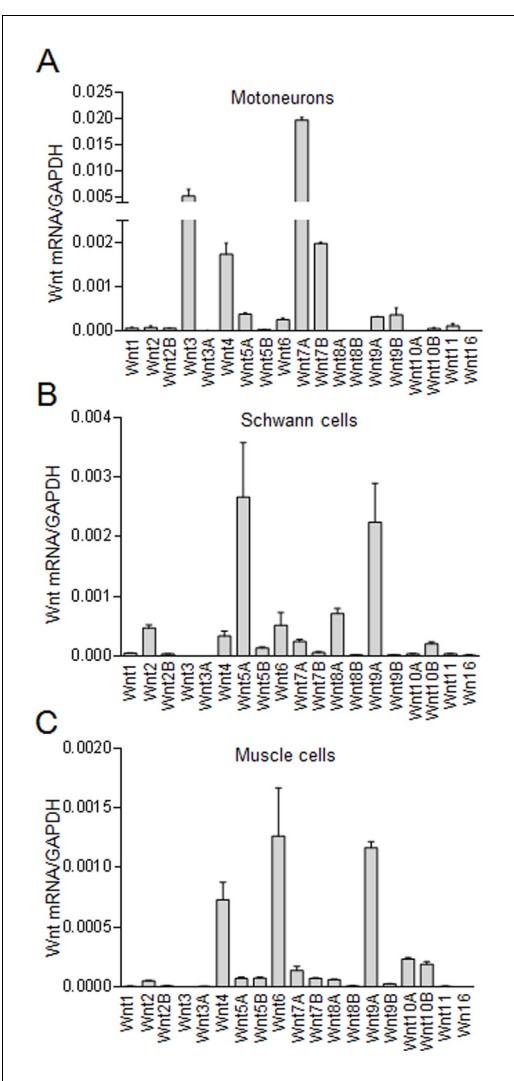

**Figure 1.** Wnt expression in motoneurons, Schwann cells, and muscle cells. (**A**) Wnt expression in motoneurons. Motoneurons in HB9-tdTomato mice (P0) were isolated by FACS. mRNAs were subjected to real-time PCR with GAPDH as internal control. (**B**) Wnt expression in primary Schwann cells. Sciatic nerves were isolated from P3 mice for primary Schwann cell culture. mRNAs were subjected to real-time PCR with GAPDH as internal control. (**C**) Wnt expression in C2C12 muscle cells. mRNAs were isolated from C2C12 myotubes and subjected to real-time PCR with GAPDH as internal control.

DOI: https://doi.org/10.7554/eLife.34625.002

The following figure supplement is available for figure 1:

**Figure supplement 1.** Frizzled expression in spinal ventral horn, Schwann cells, and muscle cells.
DOI: https://doi.org/10.7554/eLife.34625.003

secreted (*Bänziger et al., 2006*; *Bartscherer et al., 2006*; *Goodman et al., 2006*). Wls null mouse embryos exhibit defects in establishment of the body axis (*Fu et al., 2009*). We generated mutant mice that lacked Wls specifically in motorneurons, skeletal muscles, or Schwann cells and characterized NMJ development by using a combination of morphological and functional approaches. We have also identified the Wnt ligands whose secretion requires Wls and determined whether the NMJ deficits of HB9-Wls$^{-/-}$ mice could be ameliorated by supplying candidate Wnts. Our results support a model where motoneurons secret Wnts such as Wnt7A and Wnt7B for nerve terminal differentiation, suggesting dysregulation of Wnt signaling as a pathological mechanism of neuromuscular disorders.

## Results

### Unique Wnt expression patterns in motoneurons, Schwann cells and muscles

The NMJ is a tripartite synapse consisting of muscle fibers, Schwann cells, and motoneuron terminals (*Darabid et al., 2014*; *Wu et al., 2010*). We determined Wnt expression patterns in these cells. To isolate motoneurons, spinal cords were dissected out from P0 HB9-tdTomato mice where motoneurons were labeled by tdTomato (*Figure 1A*). Dissociated motoneurons were subjected to FACS analysis and RNA extraction. Real-time PCR analysis identified many Wnts in motoneurons. Noticeably, Wnt7A appeared to be the most abundant, followed by Wnt3, Wnt4, and Wnt7B. Expression of other Wnts appeared to be low in motoneurons. Dominant Wnts in Schwann cells appeared to be Wnt5A and Wnt9A whereas Wnt4, Wnt6 and Wnt9A were the most abundant in muscle cells (*Figure 1B and C*).

### No apparent NMJ deficits by Wls mutation in Schwann cells or muscles

We determined whether mutation of Wls in muscles, Schwann cells, or motoneurons alters the NMJ formation. HSA::Cre mice express Cre specifically in myotomal regions of somites beginning at E9.5, and Cre protein is detectable in almost all skeletal muscle fibers in P0 mice (*Figure 2—figure supplement 1A*) (*Escher et al., 2005*; *Li et al., 2008*). Cre in Wnt1::Cre mice is expressed in neural crest cells and their derivatives including Schwann cells (*Figure 2—figure supplement 1E*) (*Fu et al., 2011*). We crossed Wls$^{f/f}$ mice with HSA::Cre and Wnt1::Cre mice, respectively. Resulting HSA-Wls$^{-/-}$ mice and Wnt1-Wls$^{-/-}$ mice were produced at expected ratios following the Mendel rule with comparable body size to control (HSA::Cre or Wls$^{f/f}$) mice. Western blot showed that, most, if not all of Wls proteins were lost in skeletal muscle homogenates in HSA-Wls$^{-/-}$ mice (*Figure 2—figure supplement 1B*). To characterize NMJs, muscles were stained whole-mount with antibodies against neurofilament (anti-NF) and synaptophysin (anti-Syn) to label axons and nerve terminal synaptic vesicles, and rhodamine-conjugated α-bungarotoxin (R-BTX) to visualize postsynaptic AChR clusters. HSA-Wls$^{-/-}$ mice appeared to be normal at birth without apparent deficits in AChR clusters (number and location), motor nerve terminals (arborization) and innervation of AChR clusters (*Figure 2—figure supplement 1C and D*). These results were in agreement with a previous report (*Remédio et al., 2016*) and indicate that muscle Wls is dispensable for NMJ formation.

Western blotting of sciatic nerve homogenates revealed a reduction of Wls protein in Wnt1-Wls$^{-/-}$ mice (*Figure 2—figure supplement 1F*). The residual level of Wls protein was likely to be from nerve axons. Wnt1-Wls$^{-/-}$ mice died soon after birth with craniofacial skeleton defects, in accord with the notion that Wnt signaling in neural crests is critical for craniofacial skeleton development (*Fu et al., 2011*). However, NMJs in Wnt1-Wls$^{-/-}$ mice were similar to those of control mice in terms of number, location, innervation of AChR clusters and motor nerve arborization (*Figure 2—figure supplement 1G and H*). These results suggest that Wls in Schwann cells is not necessary for NMJ formation.

### Reduced body weight and muscle strength in Wls motoneuron knockout mice

To determine whether Wnts from motoneurons are involved in NMJ formation, we crossed HB9::Cre mice with Wls$^{f/f}$ mice. HB9 is a motoneuron-specific transcription factor that begins to express at E9.5 and is critical for motoneuron differentiation (*Arber et al., 1999*; *Thaler et al., 1999*). Cre expression in HB9::Cre mice was verified by crossing with Rosa-tdTomato reporter mice. As shown in *Figure 2A*, the Tomato fluorescence protein was specifically restricted in motor nerves that were

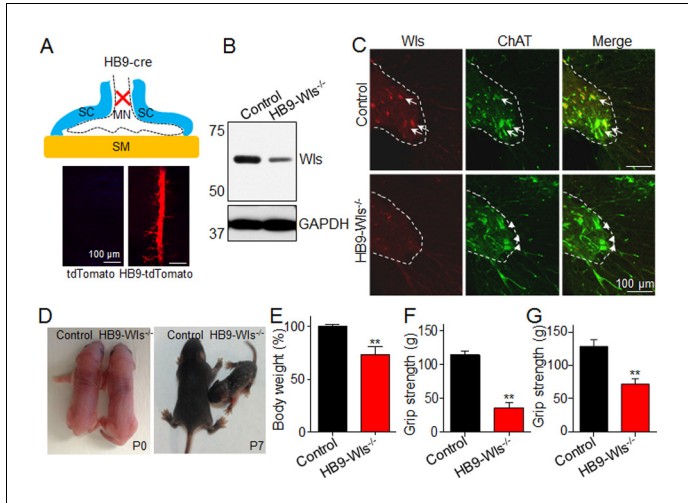

**Figure 2.** Ablation of Wls expression in motoneuron causes muscle weakness. (**A**) Up: NMJ structure. Cre targets motoneuron (HB9::cre). Down: Cre expression was verified by crossing HB9::cre mice with Rosa-tdTomato reporter mice (HB9-tdTomato). HB9::cre drives tdTomato expression in the middle line of diaphragm where motoneuron innervates in the muscle (HB9-tdTamato, (P7). MN, motoneuron; SC, Schwann cells; SM, skeletal muscle. (**B**) Immunoblot of Wls in sciatic nerve from 2-month-old control and HB9-Wls[-/-] mutant. Notice that the reduced Wls protein expression in the mutant. (**C**) Immunostaining of Wls in spinal cord from 2-month-old control and HB9-Wls[-/-] mutant. Notice that Wls staining signal is lost in the ChAT positive cells in the spinal ventral horn in the HB9-Wls[-/-]. Anti-Wls (Red); Anti-ChAT (Green). Arrows indicate colocalization of Wls and ChAT protein. (**D**) Two genotypes at P0 and P7. Notice that apparent normal size at P0, and reduced body size at P7. (**E**) Body weight of two genotypes at P7. Control, n = 20; mutant, n = 11; Unpaired t-test, **p<0.01. (**F**) Grip strength of two genotypes at P60. Control, n = 5; mutant, n = 5. Unpaired t-test, **p<0.01. (**G**) Grip strength of two genotypes with same body weight at P60. Control, n = 5; mutant, n = 5. Unpaired t-test, **p<0.01.

DOI: https://doi.org/10.7554/eLife.34625.004

The following figure supplements are available for figure 2:

**Figure supplement 1.** Ablation of Wls in skeletal muscle or Schwann cell has no apparent NMJ morphological defects.
DOI: https://doi.org/10.7554/eLife.34625.005

**Figure supplement 2.** NMJ formation appears grossly normal in HB9-Wls[-/-] mice.
DOI: https://doi.org/10.7554/eLife.34625.006

**Figure supplement 3.** Comparable NMJ formation between HSA-Wls[-/-] and HB9/HSA-Wls[-/-], Wnt1-Wls[-/-] and HB9/Wnt1-Wls[-/-] mice.
DOI: https://doi.org/10.7554/eLife.34625.007

located in the middle of diaphragm in HB9-tdTomato mice, but not in control mice (**Figure 2A**). Western blotting of sciatic nerve homogenates revealed a reduction of Wls protein (**Figure 2B**). The residual Wls was likely to due to Wls from sensory axons and Schwann cells. To determine that Wls was truly ablated from motoneurons, we stained spinal cord sections with anti-Wls antibody. As shown in **Figure 2C**, Wls was readily detectable in ChAT positive motoneurons in ventral horns of the spinal cord of control mice, but not those of HB9-Wls[-/-] mice. These results indicate that Wls was ablated from motoneurons in HB9-Wls[-/-] mice.

HB9-Wls[-/-] pups were born at expected Mendelian ratio and were similar in body size at birth, compared to control pups. However, as mice developed, HB9-Wls[-/-] pups showed growth retardation with smaller body sizes (100.0 ± 2.2% in control vs 68.6 ± 9.4% in HB9-Wls[-/-], p<0.01, **Figure 2D and E**). One third of HB9-Wls[-/-] mice died prematurely before adulthood. At 2 months old, the body weight of surviving HB9-Wls[-/-] mice was 31% less than control. In particular, grip strength was reduced by 69%, compared with control (113.4 ± 5.8 g in control mice to 35.2 ± 7.9 g in HB9-Wls[-/-] mice; p<0.01) (**Figure 2F**). Because body weights were reduced in HB9-Wls[-/-] mice, we also compared grip strength between two groups of mice with similar body weights (29.3 ± 1.1 g vs 27.7 ± 1.6 g for control and HB9-Wls[-/-], respectively, p>0.05). As shown in **Figure 2G**, HB9-Wls[-/-]

mice had reduced grip strength (129 ± 9.7 g in control mice; 71.8 ± 7.9 g in HB9-Wls$^{-/-}$ mice, p<0.01). These results indicate that motoneuron Wls is required for development of body weight and muscle strength.

## Age-dependent nerve terminal swellings and poor innervation

To determine whether loss of Wls in motoneurons alters the NMJ formation, we first characterized NMJs of mouse embryos (E18.5). As shown in suppl *Figure 2A*, the primary branches of phrenic nerves are localized in central regions of muscle fibers; the numbers of secondary and tertiary branches were similar between control and HB9-Wls$^{-/-}$ mice. The morphology of nerve terminals was comparable between the two genotypes and each of them was associated with an AChR cluster (*Figure 2—figure supplement 2A*). These results indicate that nerve terminal pathfinding to the middle region of muscle fibers and defasciculation or arborization appears grossly normal. Postsynaptically, AChR clusters appeared as opaque ovals and were enriched in the middle region of muscle fibers, outlining a central band in control mice (*Wu et al., 2010*) (*Figure 2—figure supplement 2A*). The width of the central region that occupied by AChR clusters was similar in control and in HB9-Wls$^{-/-}$ diaphragms (172 ± 25.4 μm vs 198 ± 25.5 μm, p>0.05). The density and size of AChR clusters in HB9-Wls$^{-/-}$ mice were also similar to those in control mice (AChR size: 100 ± 5.0% vs 87.9 ± 4.3%, p>0.05; AChR density: 537 ± 28.0 vs 556 ± 22.2 per μm$^2$, p>0.05) (*Figure 2—figure supplement 2B–2D*). These results suggest that HB9-Wls$^{-/-}$ mice were able to form the NMJ in the absence of motoneuron Wls, in agreement with the finding of normal body weights of P0 HB9-Wls$^{-/-}$ mice.

Normal NMJs at P0 pups were unable to explain muscle weakness at age of 2 months. Therefore, we isolated gastrocnemius from 2-month-old mice and analyzed NMJ morphology by whole-mount staining. To avoid potential bias during imaging process, NMJs were viewed from three different axes and only largest images of AChR clusters were captured for analysis (*Shen et al., 2013*). Mature NMJs had well developed into characteristic pretzel-like structures with complex continuous arrays that were innervated by motor nerve terminals (*Shen et al., 2013*) (*Figure 3A*). The arrays of HB9-Wls$^{-/-}$ mice appeared to be as complex as control mice, but less continuous (*Figure 3A and B*). There were small pieces of AChR clusters (arrow, *Figure 3A and B*), and the number of AChR cluster fragments were increased to 3.0 ± 0.3 fragments per NMJ in HB9-Wls$^{-/-}$ mice from 1.4 ± 0.2 fragments per NMJ in control mice (p<0.01) (*Figure 3A–3C*).

A characteristic abnormality of HB9-Wls$^{-/-}$ mice was the appearance of abnormal swellings of nerve terminals. The swellings appeared to be at axon terminals (arrowhead, *Figure 3A and B*) although we could not exclude the possibility that the swellings may bulge from axons en passant to form a different NMJ (arrowhead, *Figure 3A*). Axonal bulbs or shedding were observed during synapse elimination (*Tapia et al., 2012*). We did not find the excessive NMJ number in each single muscle fiber at different ages of HB9-Wls$^{-/-}$ mice, suggesting the normal synapse elimination in HB9-Wls$^{-/-}$ mice (*Figure 3—figure supplement 1*). Unlike axonal shedding which was observed in a small number of NMJs, axonal swellings in HB9-Wls$^{-/-}$ mice were observed in 53.5 ± 3.2% of nerve terminals (*Figure 3E*). Some swellings were co-stained with dispersed AChR signal, suggesting a problem of mutant axons in inducing AChR clusters. In addition to axonal swellings, nerve terminals in HB9-Wls$^{-/-}$ mice became detached from AChR arrays, causing misalignment between nerve terminals and AChR clusters (blue arrowhead, *Figure 3B*). Consequently, the area of AChR arrays that were covered by nerve terminals was reduced in HB9-Wls$^{-/-}$ mice (*Figure 3D*) (73.6 ± 7.1% in HB9-Wls$^{-/-}$ vs 100 ± 3.7% in control, p<0.05).

To further characterize axonal swellings, we utilized Rosa-tdTomato reporter mice where tdTomato expression is controlled by a loxP-stop-loxP cassette. When crossed with HB9-Wls$^{-/-}$ mice, HB9::Cre thus drove the expression of tdTomato and at the same time mutated the *Wls* gene in motoneurons. In diaphragms of control mice, tdTomato-labeled axons were transparent in secondary and tertiary branches, and each terminal ended with a pretzel-like enlargement. However, in HB9-Wls$^{-/-}$ mice, in addition to these transparent axon branches and terminals, many branches appeared to contain aggregates. Axon terminals in HB9-Wls$^{-/-}$ mice appeared as individual swellings without characteristic pretzel-like structure (*Figure 3F*). Quantitatively, 53% of axon terminals lost the pretzel-like structure (*Figure 3F*), in agreement with the result from immunostaining (*Figure 3A and B*). Together, these morphological results demonstrate that motoneuron-specific Wls mutant mice were able to form NMJs, but displayed axonal deficits at adult, indicating an age-dependent requirement of Wls in NMJ maintenance.

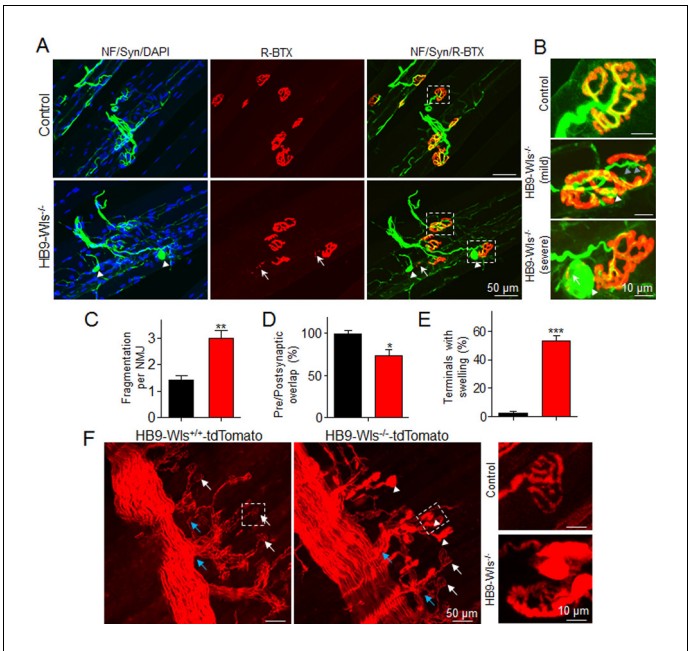

**Figure 3.** Abnormal pre- and postsynaptic NMJ structure in HB9-Wls$^{-/-}$ mutant mice. (**A**) Immunostaining of NMJs in gastrocnemius of 2-month-old control and HB9-Wls$^{-/-}$ mice with anti-NF/anti-Syn (Green) and R-BTX (Red). Nuclei were labeled with DAPI (Blue). In mutant, some nerve terminals are swollen (white arrowhead) and AChR clusters are broken (white arrow). (**B**) The high magnification of boxed area in A. White arrowhead indicates swollen nerve terminals. Blue arrowhead indicates motor nerve detaching from AChR clusters; White arrow indicates broken pieces of AChR. (**C**) Statistical results of AChR fragmentation per NMJ. n = 3 mice per group; Unpaired t-test, **p<0.01. (**D**) Percentage of AChR cluster innervated with nerve terminals. n = 3 mice per group; Unpaired t-test, *p<0.05. (**E**) Percentage of nerve terminal swellings. n = 3 mice per group; Unpaired t-test, ***p<0.001. (**F**) The diaphragm of 2-month-old HB9-tdTomato and HB9-WLS$^{-/-}$ tdTomato mice. Nerve terminal swellings are indicated with white arrowhead. Normal nerve terminals are indicated with white arrow. Axon branch are indicated with blue arrow. Boxes indicated by dash line were enlarged in the right.
DOI: https://doi.org/10.7554/eLife.34625.008

The following figure supplement is available for figure 3:

**Figure supplement 1.** Wls loss in motoneuron does not enhance synapse number in single muscle fiber.
DOI: https://doi.org/10.7554/eLife.34625.009

## Reduced synaptic vesicles, junctional folds and increased synaptic cleft width

To further characterize NMJ defects, we performed EM analysis. In control NMJs, nerve terminals are filled with synaptic vesicles and covered with processes of terminal Schwann cells. Membranes of muscle fibers opposing axon terminals invaginate to form junctional folds where AChRs are enriched (*Figure 4A*). These unique structures ensure the neurotransmitter ACh, released from axonal terminals, to be efficiently captured by AChRs on the crest of junctional folds (*Li et al., 2018*; *Sanes and Lichtman, 2001*; *Wu et al., 2010*). However, these NMJ structures were largely distorted in HB9-Wls$^{-/-}$ mice, which made it difficult to identify characteristic NMJs. Noticeably, the number of synaptic vesicles was reduced in nerve terminals (107 ± 14.6 per μm$^2$ in control vs 25.6 ± 3.0 in HB9-Wls$^{-/-}$, p<0.01, black arrowhead) (*Figure 4A and B*). The number of junctional folds was reduced (1.2 ± 0.3 in HB9-Wls$^{-/-}$ vs 3.0 ± 0.5 in control per μm, p<0.05, black arrow) (*Figure 4*). Synaptic cleft was uneven in HB9-Wls$^{-/-}$ mice and appeared to be larger than that of control mice (2.1 ± 0.3 folds in HB9-Wls$^{-/-}$ mice, compared with that in control mice, p<0.01) (*Figure 4A and D*). In control NMJs, basal lamina was visible in synaptic cleft and between junctional folds, with highest electron density in the middle. However, in HB9-Wls$^{-/-}$ mice, basal lamina density was reduced without a high-density midline (empty arrowheads, *Figure 4A*). These EM data are in accord with those from light

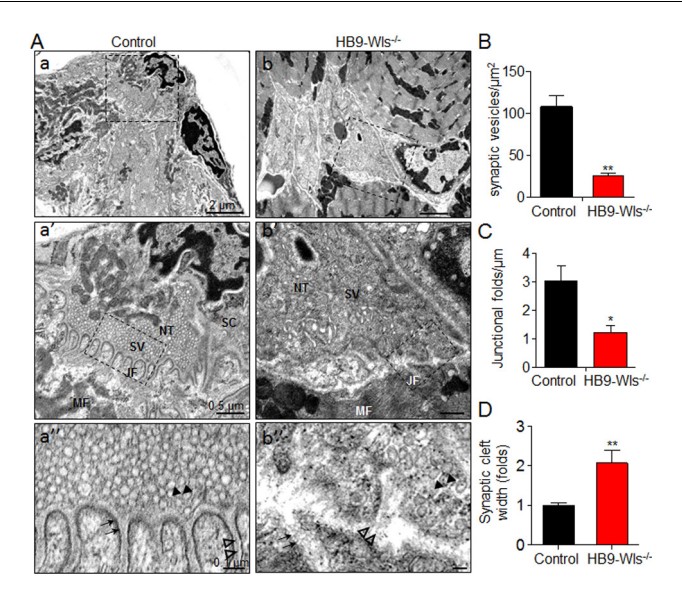

**Figure 4.** Disruption of NMJ ultrastructure in HB9-Wls$^{-/-}$ mutant mice. (**A**) Representative EM images of diaphragm NMJs in P15 control (**a, a', a''**) and HB9-Wls$^{-/-}$ mice (**b, b', b''**). Notice that there is a dramatic reduction of synaptic vesicles number in the HB9-Wls$^{-/-}$ (black arrowhead, (**a'' and b''**) and loss of synaptic junctional folds (black arrow, (**a'' and b''**) and basal lamina (empty arrowhead, (**a'' and b''**) in HB9-Wls$^{-/-}$. Boxed regions are shown at higher magnification immediately below. NT, nerve terminal; MF, muscle fiber; SC, Schwann cell; SV, synaptic vesicle; JF, junctional fold. Scale bars: 2.0 μm (top); 0.5 μm (middle); 0.1 μm (bottom). (**B**) Synaptic vesicle density in two genotypes. n = 3 mice per group; Unpaired t-test, **p<0.01. (**C**) Number of synaptic junctional folds. n = 3 mice per group; Unpaired t-test, *p<0.05. (**D**) Synaptic craft width. n = 3 mice per group; Unpaired t-test, **p<0.01.
DOI: https://doi.org/10.7554/eLife.34625.010

microscopic studies (*Figure 3*) and demonstrate that Wls in motoneurons is critical for the integrity of NMJ structure.

## Impaired neuromuscular transmission in HB9-Wls$^{-/-}$ mice

To investigate pathophysiological mechanisms of muscle weakness in HB9-Wls$^{-/-}$ mice, we measured compound muscle action potentials (CMAPs) to determine if there is impaired neuromuscular transmission. Gastrocnemius was recorded in response to repetitive nerve stimuli (*Shen et al., 2013*). In control mice, CMAPs showed little change after 10 consecutive nerve stimuli at frequencies from 2 Hz to 40 Hz (*Figure 5A and B*). While in HB9-Wls$^{-/-}$ mice, CMAPs recorded at 40 Hz began to decrease at the second stimulus and significantly decreased from the fourth; the decrement of CMAP at the tenth stimulus was about 10% (*Figure 5B*). The reduction of CMAPs in HB9-Wls$^{-/-}$ mice was frequency-dependent (*Figure 5C*), which indicates progressive loss of successful neuromuscular transmission after repeated stimulation.

To further investigate whether the neurotransmission deficits result from pre- and/or postsynaptic impairment, we recorded miniature endplate potentials (mEPPs), events generated by spontaneous vesicle release, in the single muscle fiber of hemidiaphragms of P30 HB9-Wls$^{-/-}$ mice. As shown in *Figure 5D and E*, mEPP amplitudes were reduced in HB9-Wls$^{-/-}$ mice, compared with controls (1.20 ± 0.08 mV in control vs 0.76 ± 0.06 mV in HB9-Wls$^{-/-}$; p<0.01), indicative of reduced AChR density. mEPP frequencies were also decreased by ~40% in HB9-Wls$^{-/-}$ diaphragms, compared with control (0.78 ± 0.03 Hz in HB9-Wls$^{-/-}$ vs 1.20 ± 0.08 Hz in control, p<0.01, *Figure 5D and F*). This result demonstrates deficits in spontaneous ACh release from motor nerve terminals, consistent with morphological presynaptic defects in mice lacking Wls in motoneurons. We observed similar electrophysiological defects at P15 HB9-Wls$^{-/-}$ mice (*Figure 5—figure supplement 1*). The result was consistent with that showed by EM, in which postsynaptic as well as presynaptic membranes were disorganized (*Figure 4*).

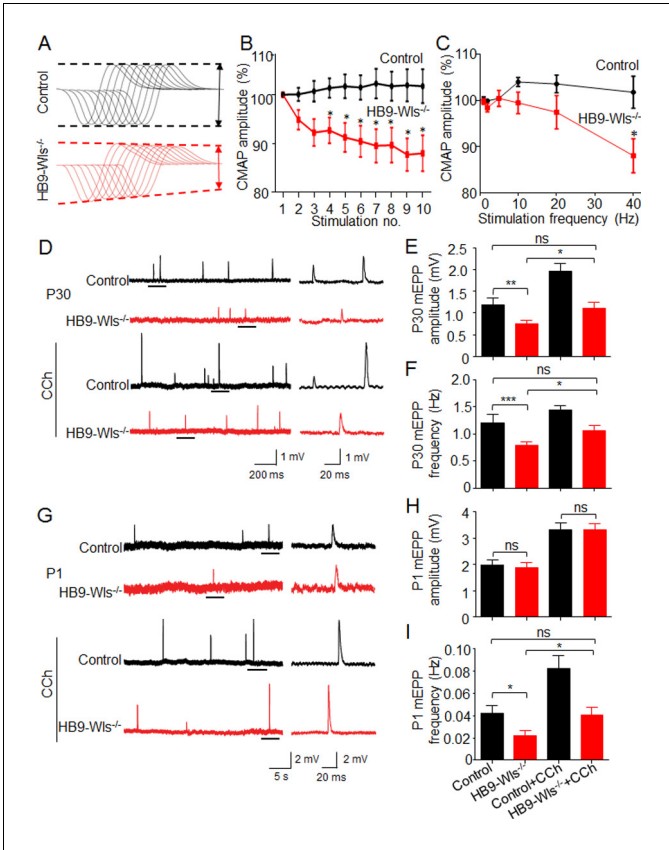

**Figure 5.** Wls loss in motoneuron impairs neurotransmission initially from presynapse. (**A**) CMAPs were recorded in gastrocnemius from P30 mice in response to a train of 10 submaximal stimuli at different frequencies. The first stimulus response in control mice was designated as 100%. Representative 10 CMAP traces, shown stacked in succession for better comparison. (**B**) Reduced CMAP amplitudes at 40 Hz in HB9-Wls$^{-/-}$ mice. n = 4 mice per group. Two-way ANOVA, *p<0.05. (**C**) CMAP amplitudes of the tenth stimulation at different stimulation frequencies. n = 4 mice per group. Two-way ANOVA, *p<0.05. (**D**) Representative mEPP traces from P30 mice. mEPPs were recorded from hemidiaphragms. Traces underlined on the left are enlarged on the right. (**E**) Reduced mEPP amplitudes in mutant mice and CCh effects at P30. $F_{(2, 9)}$=13.56, ns, p>0.05, **p<0.01. n = 4 mice per group, 5–6 muscle fibers per mouse; One-way ANOVA. (**F**) Reduced mEPP frequencies in mutant mice and CCh effect at P30. $F_{(2, 9)}$=14.2, ns, p>0.05, **p<0.01, *p<0.05, n = 4 mice per group, 5–6 muscle fibers per mouse; One-way ANOVA. (**G**) Representative mEPP traces from P1 mice. (**H**) Comparable mEPP amplitude between two genotypes and CCh effect at P1. $F_{(2, 9)}$=63.07, ns, p>0.05, ***p<0.001. n = 4 mice per group, 5–6 muscle fibers per mouse; One-way ANOVA. (**I**) Reduced mEPP frequency in mutant mice and CCh effect at P1. $F_{(2, 9)}$=12.87, ns, p>0.05, **p<0.01. n = 4 mice per group, 5–6 muscle fibers per mouse; One-way ANOVA.

DOI: https://doi.org/10.7554/eLife.34625.011

The following figure supplement is available for figure 5:

**Figure supplement 1.** Impairment of pre- and postsynaptic prosperity in P15 HB9-Wls$^{-/-}$ mice.

DOI: https://doi.org/10.7554/eLife.34625.012

Because Wls was specifically mutated in motoneurons, it was interesting that HB9-Wls$^{-/-}$ mice at P30 and P15 displayed postsynaptic deficits. Such deficits may result as a consequence of deteriorated presynaptic terminals. This hypothesis would predict that mutant mice at earlier age display only presynaptic, but not postsynaptic deficits. To test this hypothesis, mEPPs were measured in diaphragm at neonatal pups (P1) when the NMJ morphology appeared normal. mEPP frequency in HB9-Wls$^{-/-}$ NMJs was reduced by 42% (0.042 ± 0.003 Hz in control vs 0.022 ± 0.002 Hz in mutant, p<0.01) (**Figure 5G and I**), indicative of presynaptic deficits. In contrast, there was no significant difference in mEPP amplitudes between control and HB9-Wls$^{-/-}$ mice (1.98 ± 0.10 mV in control vs 1.88 ± 0.09 mV in HB9-Wls$^{-/-}$, p>0.05) (**Figure 5G and H**). These functional results demonstrate that

presynaptic deficits occurred in advance of postsynaptic deficits, agreeing the notion that EM and electrophysiological analysis are more sensitive in revealing NMJ deficits than light microscope analysis.

To further test this model, we studied effects of carbochol (CCh), an ACh agonist, on neurotransmission in HB9-Wls$^{-/-}$ mice. CCh (10 μM) increased both mEPP frequency and amplitude at P1 and P30 mice (*Figure 5D–5I*), in agreement with previous reports (*Katz and Miledi, 1972*; *Miyamoto and Volle, 1974*). Interestingly, CCh also elevated mEPP frequency and amplitude in Wls mutant diaphragms to similar or higher levels of controls in the absence of CCh (*Figure 5D–5I*). Nevertheless, mEPP frequency and amplitude in the presence of CCh were lower in mutant, compared with control at age of P30. These results indicate that transmission impairment in mutant mice could be attenuated by ACh agonist.

## SV2-positive aggregates in HB9-Wls$^{-/-}$ axons

To determine molecular components of axon swellings, muscles were stained with antibody against NF. As shown in *Figure 6A*, the swellings were positive for NF, indicating that they may contain neurofilaments. They could also be stained with antibody against S100, a marker of Schwann cells, suggesting that the swellings were covered by Schwann cells (*Figure 6A*).

Both EM and electrophysiological analyses revealed presynaptic deficits. In particular, vesicle release was reduced and there were fewer synaptic vesicles in HB9-Wls$^{-/-}$ mice, compared with control mice (*Figures 4* and *5*). We posited that it may result from a problem in vesicle transportation to nerve terminals. To test this hypothesis, gastrocnemius of 2-month-old mice was immunostained with antibody against SV2, a marker of synaptic vesicles. In control NMJs, SV2-positive vesicles were enriched at NMJs and showed puncta-like structures, which were distributed evenly in arrays of AChR clusters (*Figure 6B*). In HB9-Wls$^{-/-}$ NMJs, SV2 staining signal was spotty and discontinuous, faint or missing in many areas where AChR staining was strong. Noticeably, SV2-positive aggregates were seen in motoneuron axons in HB9-Wls$^{-/-}$ mice (white arrow, *Figure 6B*). Such SV2 aggregates were not observed in control mice. These results suggest abnormal accumulation of synaptic vesicles in axons. To test this further, we isolated phrenic nerves which contains purer motoneuron axons than sciatic nerve and performed immunoblotting to check the protein levels of synaptic vesicles. As

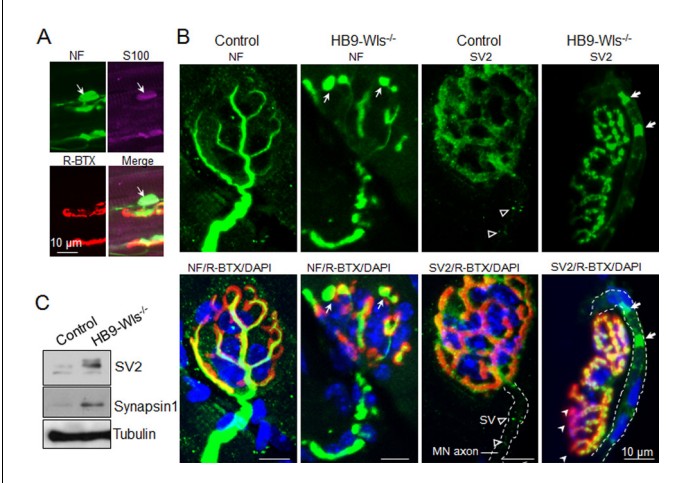

**Figure 6.** Clustered synaptic vesicles in HB9-Wls$^{-/-}$ axons. (**A**) Immunostaining of axonal terminal swelling in gastrocnemius of P21 HB9-Wls$^{-/-}$ mice with anti-NF (Green), anti-S100 (Purple), R-BTX (Red). (**B**) Immunostaining of NMJs in gastrocnemius of 2-month-old control and mutant mice with anti-NF or anti-SV2 to label synaptic vesicles (Green). R-BTX indicates the AChR (Red). Cell nucleus is labeled with DAPI (Blue). Dashline indicates motor axon. Notice that in the mutant NMJ, there are aberrant deposits of synaptic vesicles in the motor axon (bold white arrow in mutant and empty arrowhead in control), and reduced overlap between pre and post-synapse (white arrowhead). Nucleuses around motor axons are from Schwann cells. (**C**) Immunoblot of SV2 and Synapsin 1 in phrenic nerve at 2-month-old control and mutant mice.

DOI: https://doi.org/10.7554/eLife.34625.013

shown in *Figure 6C*, compared with control, SV2 and Synapsin 1 protein level in phrenic nerve was increased in HB9-Wls$^{-/-}$ mutant mice. Together, these results identify a cellular mechanism for reduced synaptic vesicles at NMJs in mice lacking Wls in motoneurons.

## Motoneuron Wnts for nerve terminal development

Wls is critical for sorting and transporting Wnts from the endoplasmic reticulum to the cell surface (*Yang et al., 2008*). NMJ deficits in mutant mice lacking Wls in motoneurons suggest a role of Wnts from motoneurons for NMJ development. Motoneurons express many Wnts as shown in *Figure 1A*. We attempted to identify which Wnts were vulnerable to Wls deficiency. We co-transfected Wls-shRNA that reduced Wls expression (data not shown) and individual Wnt constructs in HEK293 cells and analyzed Wnts level in condition medium. Each Wnt ligand was tagged with a Flag epitope. We focused on Wnts that were relatively abundant in motoneurons. As shown in *Figure 7A and B*, Wls knockdown reduced most of Wnts level including Wnt5A, Wnt7A, Wnt7B, Wnt9A, and Wnt9B in the medium, in agreement with previous reports.

The observations that Wnt5A, Wnt7A, Wnt7B, and Wnt9B were expressed in the spinal cord (*Figure 1A*) and their levels in the medium were largely reduced by Wls knockdown (*Figure 7A and B*) suggest that these Wnts may be utilized by motoneurons to regulate the NMJ formation. To test this hypothesis, we determined whether Wnt5A, Wnt7A, Wnt7B, and Wnt9B recombinant proteins were able to diminish NMJ deficits in HB9-Wls$^{-/-}$ mice. TA muscles of mutant pups were injected with respective Wnts or vehicle (as control) 10 times, once every three days for a period of 30 days.

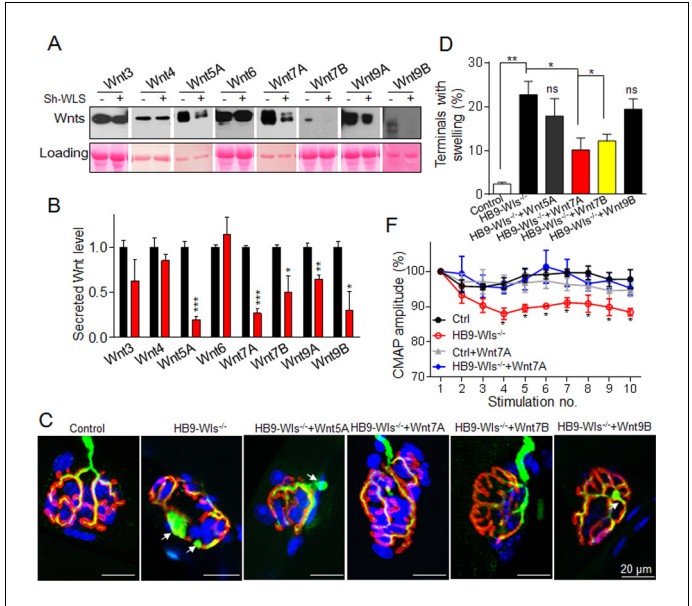

**Figure 7.** Wnt7A partially rescues axonal terminal swelling in HB9-Wls$^{-/-}$ mice. (**A**) Wls-dependent Wnts secretion. HEK293T cells were co-transfected with shRNA of Wls and motoneuron Wnts cDNA with a Flag tag. Conditioned medium was collected for testing secreted Wnt. Ponceau S red indicates the loading. (**B**) Statistical results of secreted Wnt protein levels in conditioned medium in A. *p<0.05, **p<0.01, ***p<0.001. Unpaired t-test. Three independent experiments were performed. (**C**) Immunostaining of mutant TA muscles injected with Wnt recombinant proteins. Wnt recombinant proteins (10 ng/µl, 20 µl) were injected into TA muscles of right leg in HB9-Wls$^{-/-}$ mice at every three days. TA muscle of left leg in the same mouse was injected with vesicle as control. Thirty days later, TA muscle was isolated for NMJ staining. Compared with control, Wnt5A or Wnt9B, Wnt7A and Wnt7B partially rescued the axon terminal swelling in HB9-Wls$^{-/-}$ mice. Anti-NF/Syn: Green; R-BTX: Red; DAPI: Blue. (**D**) Statistical results of F. Control group: n = 5 mice; HB9-Wls$^{-/-}$ group: n = 6 mice; HB9-Wls$^{-/-}$ + Wnt5A group: n = 6 mice; HB9-Wls$^{-/-}$ + Wnt7A group: n = 5 mice. HB9-Wls$^{-/-}$ + Wnt7B group: n = 5 mice. HB9-Wls$^{-/-}$ + Wnt9B group: n = 5 mice. Unpaired t-test, *p<0.05, **p<0.01. ns, non-significant. (**E**) Increased CMAP amplitudes in HB9-Wls$^{-/-}$ mice by Wnt7A at 30 Hz stimulation. Control group: Black; HB9-Wls$^{-/-}$ group: Red; Control +Wnt7A group: Gray; HB9-Wls$^{-/-}$ + Wnt7A group: Blue. n = 4 mice per group; Two-way ANOVA, *p<0.05.

DOI: https://doi.org/10.7554/eLife.34625.014

Remarkably, muscles injected with Wnt7A and Wnt7B displayed fewer axon swellings, compared with muscles injected with vehicle (*Figure 7C and D*). This effect appeared to be specific because muscles injected Wnt5A or Wnt9B displayed similar number of axon swellings (NMJs with swellings: $22.7 \pm 3.0\%$ in HB9-Wls$^{-/-}$; $17.9 \pm 3.9\%$ in HB9-Wls$^{-/-}$ + Wnt5A; $10.1 \pm 2.7\%$ in HB9-Wls$^{-/-}$ + Wnt7A; $12.2 \pm 1.5$ in HB9-Wls$^{-/-}$ + Wnt7B; $19.4 \pm 2.4\%$ in HB9-Wls$^{-/-}$ + Wnt9B). To further test this, we determined whether neuromuscular transmission impairment in HB9-Wls$^{-/-}$ mice could be attenuated by Wnt7A administration. As shown in *Figure 7E*, CMAPs of HB9-Wls$^{-/-}$ mice were resued by Wnt7A administration. Together, these results support a model where motoneurons release Wnts to regulate NMJ development.

## Discussion

The roles of Wnt in NMJ presynaptic differentiation remain elusive. Here we found that blocking Wnt secretion from motoneurons causes NMJ degeneration evidenced by muscle weakness, impaired neurotransmission, distorted NMJ structure, motor axonal terminal swelling, and synaptic vesicle reduction. Rescue experiments identified Wnt7A and Wnt7B as key Wnts secreted from motoneurons for NMJ development. Our results provide a novel role of Wnt ligand secretion from motoneurons in NMJ development and maintenance.

NMJ deficits were observed in mice lacking Wls in motoneurons. The 19 Wnts have different expression patterns in motoneurons, Schwann cells, and muscles (*Figure 1*). Dominant Wnts in motoneurons are Wnt7A, Wnt3, Wnt7B, and Wnt4 (*Figure 1*), of which Wnt7A and Wnt7B are sensitive to Wls regulation (*Figure 7*). In addition, these two Wnts are able to diminish NMJ deficits in HB9-Wls$^{-/-}$ mice (*Figure 7*). A parsimonious explanation of these results is that motoneurons release Wnt7A and Wnt7B to regulate NMJ development. Knockout Wls in Schwann cells or muscles alone or together with motoneurons seemed to have little effect on NMJ morphology (*Figure 2—figure supplements 1–3*). In agreement, no apparent NMJ deficits were detected when Wls was knocked out in muscles by Pax3::Cre (*Remédio et al., 2016*). These results could suggest that motoneuron Wnts cannot be compensated by Wnts from skeletal muscle fibers or Schwann cells. A more detailed analysis of NMJs in the triple knockout mice will help to dissect whether motoneuron Wnts compensate Wnts from Schwann cells or muscles during NMJ formation.

How does motoneuron Wnts regulate NMJ development? In vitro, Wnt7A was shown to inhibit agrin-induced AChR clustering (*Barik et al., 2014b*). However, we did not observe a change in AChR cluster size or intensity after Wnt7A injection (data not shown). Rather, Wnt7A reduced the number of axonal swellings and functionally rescued neurotransmission impairment in the HB9-Wls$^{-/-}$, suggesting that Wnt7A mainly acts on nerve terminals in the mutant. This notion is supported following two lines of evidence. First, the unique presynaptic phenotypes of HB9-Wls$^{-/-}$ mice - swollen axon terminals and enlarged varicosities were not observed in MuSK or Rapsyn mutant mice where AChR clusters were abolished and nerve terminals arborize extensively (without swellings, however) (*DeChiara et al., 1996*; *Gautam et al., 1995*; *Li et al., 2016*). Second, they were not observed in muscle-specific β-catenin knockout or transgenic mice overexpressing β-catenin in muscles (*Li et al., 2008*; *Liu et al., 2012*; *Wu et al., 2015b*; *Wu et al., 2012a*). Wnt7A has been reported to initiate both canonical and non-canonical pathway (*Heasley and Winn, 2008*; *Posokhova et al., 2015*; *von Maltzahn et al., 2011*; *Winn et al., 2005*; *Yang et al., 2012*). However, motoneuron-specific deletion of β-catenin does not impair NMJ development (*Li et al., 2008*; *Wu et al., 2015a*), excluding a role of canonical pathway in motoneurons. Interestingly, Wnt non-canonical JNK pathway has been reported to regulate axonal transport of vesicles in C. elegans (*Byrd et al., 2001*). Many Frizzled were detectable in the ventral horns of the spinal cord, Schwann cells, and muscle cells (*Figure 1—figure supplement 1*). Future experiments are warranted to determine whether Wnt non-canonical pathways and which wnt receptor(s) regulate NMJ development in mammals.

Presynaptic defects in HB9-Wls$^{-/-}$ mice occur prior to postsynaptic defects evidenced by the reduction of mEPP frequency, but not amplitude, in P1 mutant (*Figure 5*). The cause to postsynaptic deficits could be complex. Because Wnts can directly regulate postsynaptic AChR assembly (*Barik et al., 2014b*; *Zhang et al., 2012*), lack of Wnts from motoneurons may thus impair postsynaptic assembly. When N-box of the AChR ε-subunit promoter is mutated, this leads to AChR deficiency and severely impaired transmission at the NMJ, but postsynaptic membrane appears normal (*Nichols et al., 1999*). On the other hand, neurotransmission blockade by botulinum toxin disrupts

NMJ morphology (*Rogozhin et al., 2008*). Problems with presynaptic assembly may impair synaptic basal lamina. In fact, in mutant mice synaptic basal lamina was disorganized without a high-density midline (*Figure 4*). This could be a mechanism of loss of postsynaptic structures.

SV2 staining is progressively restricted from axon to the nerve terminal during NMJ development (*Lupa and Hall, 1989*). This is thought to indicate active vesicle transportation during neural development. In agreement, SV2 signal was low and sparse in wild type mice at two months of age (*Figure 6B*). However, SV2 and Synapsin 1 another marker of synaptic vesicle, were increased in axons of adult mutant mice by staining and Western blot analysis (*Figure 6B and C*), which is similar to the phenotype when autophagy is dysregulated in axon (*Lüningschrör et al., 2017*). This could be a reason to reduced synaptic vesicles at mutant NMJs. At the same, increased vesicle proteins in axons may be indicative of axon regeneration in the absence of Wls. In addition to increased SV2 levels, SV2-positive aggregates were detected in Wls mutants (*Figure 6B*). However, the nature of these aggregates is unknown. They may be inclusion bodies containing SV2 proteins for degradation. Nevertheless, such axon terminals swellings and varicosities are observed in NMJs recovering from Botulinum toxin poisoning (*Rogozhin et al., 2008*) and resemble those of nerve degeneration during aging (*Bhattacharya et al., 2016*; *Lüningschrör et al., 2017*; *Valdez et al., 2010*). Together, these observations are indicative of reinnervation in response to functional denervation. It would be interesting to determine whether Wnts from motoneurons regulate synapse regeneration and their loss of function precipitates the NMJ senescence.

It is worthy pointing out that the above conclusion is based on an assumption that Wls is a cargo receptor for all 19 Wnts' secretion (*Gross et al., 2012*). Apparently, this might not be the case because individual Wnt has different sensitivities to Wls deficiency (*Figure 7*). In agreement, Zebrafish Wls mutants are surprisingly normal in early development, in contrast to Wnt5B and Wnt11 mutants (*Kuan et al., 2015*). It is possible that some Wnts may be regulated in a Wls-independent manner. For example, Wls-antisense morpholino was shown to severely impair membrane localization of Wnt5B, but not Wnt11 in zebrafish (*Wu et al., 2015a*). WntD in Drosophila does not require lipidation by Porcupine and thus may be secreted independent of Wls (*Chen et al., 2012*; *Ching et al., 2008*; *Richards et al., 2014*). Wnt3A seems not responsive to Wls overexpression (*Galli et al., 2016*). It is also possible that there is another not-yet-reported Wls-like homology gene in the rodents. It would be interesting to determine whether NMJ development is regulated by Wls-insensitive and/or Wls-independent Wnts.

## Materials and methods

### Materials

Wls$^{f/f}$ mice, in which the exon 3 of *Wls* gene was flanked by loxp sites, were described previously (*Fu et al., 2011*). The following mouse lines were described in the literature: HB9::Cre (*Wu et al., 2012b*), HSA::Cre (*Wu et al., 2012b*), and Wnt1::Cre (*Fu et al., 2011*). Rosa-tdTomato (Jax, Cat# 007909) and Rosa-Laz (Jax, Cat# 003309) were purchased from Jackson Laboratory (Bar Harbor, ME). Unless otherwise indicated, control mice were either relevant floxed or Cre littermates. Mice were housed in a room with a 12 hr light/dark cycle with ad libitum access to water and rodent chow diet. Experiments with animals were approved by Institutional Animal Care and Use Committees of Augusta University, Case Western Reserve University, and Zhejiang University. HEK293T cells were purchased from ATCC (RRID:CVCL_0063) and were certified authentic and mycoplasma free.

### Grip strength measurement

Muscle strength was determined using an SR-1 hanging scale (American Weigh Scales), as described previously (*Shen et al., 2013*). Briefly, mice were held by the tail, allowed to grip a grid connected to the scale. They were gently pulled horizontally until the grip was released. Top five values will be scored.

### Light microscopy analysis

Whole-mount staining of muscles was performed as described previously (*Shen et al., 2013*; *Wu et al., 2012b*). Briefly, diaphragm or gastrocnemius were fixed in 4% paraformaldehyde (PFA) in PBS at 4°C overnight. Muscles were rinsed with PBS and incubated with 0.1 M glycine in PBS for 30

min. After being permeabilized with 0.5% Triton X-100 in PBS for 1 hr, muscles were blocked in the blocking buffer (3% BSA, 5% goat serum, and 0.5% Triton X-100 in PBS) for 2 hr, followed by incubation with primary antibodies against neurofilament (1:1000, Cell Signaling, Cat# 2837S), synaptophysin (1:1000, Dako, Cat# M7315), or SV2 (1:1000, Developmental Studies Hybridoma Bank, Cat # SV2) in the blocking buffer at 4°C overnight. Samples were washed 3 times with 0.5% Triton X-100 in PBS and incubated with a mixture of Alexa Fluor 488-conjugated secondary antibodies (1:750; Thermo Fisher Scientific, Cat# A11001 and A11008) and Alexa Fluor 594-conjugated β-bungarotoxin (BTX) (1:2000, Sigma-Aldrich, Cat# 0006) for 2 hr. After washing 3 times with 0.5% Triton X-100 in PBS, samples were flat-mounted in Vectashield mounting medium including DAPI (Vector Laboratories). Other primary antibodies for immunostaining were anti-Wls (rabbit, 1:200, home-made) (*Yu et al., 2010*) and anti-ChAT (goat, 1:200, Millipore, Cat# AB144P). Z-serial images were collected with a Zeiss confocal laser scanning microscope (LSM 510 META 3.2) and collapsed into a single image.

## Electron microscopy (EM) analysis

EM analysis was performed as described previously (*Barik et al., 2014a*). Diaphragms were fixed in 2% glutaraldehyde and 2% PFA in PBS for 1 hr and then fixed in 1% osmium tetroxide in sodium cacodylate buffer (pH 7.3) for 1 hr at 25°C. After washing 3 times with PBS, tissues were dehydrated through a series of ethanol (30%, 50%, 70%, 80%, 90%, and 100%). After 3 rinses with 100% propylene oxide, samples were embedded in plastic resin (EM-bed 812; EMSciences). Serial thick sections (1–2 μm) of tissue blocks were stained with 1% toluidine blue, cut into ultrathin sections, mounted on 200-mesh unsupported copper grids, and stained with uranyl acetate (3% in 50% methanol) and lead citrate (2.6% lead nitrate and 3.5% sodium citrate, pH 12.0). Electron micrographs were taken using a JEOL 100CXII operated at 80 KeV. Synaptic vesicles in randomly-selected areas within 500 nm from presynaptic membrane were quantified. Junctional folds and synaptic cleft width were quantified in randomly-selected areas where more than 5 consecutive junctional folds could be visibly defined.

## Electromyography

Mice were anesthetized with ketamine and xylazine mixture (100 and 10 mg/kg body weight, respectively). The stimulation needle electrode (092-DMF25-S; TECA) was inserted near the sciatic nerve in the thigh. The reference needle electrode was inserted near the Achilles tendon, and the recording needle electrode was inserted into the middle of the gastrocnemius or TA muscles. The reference and recording electrodes were connected to an Axopatch 200B amplifier (Molecular Devices). Supramaximal stimulation was applied to the sciatic nerve with trains of 10 stimuli at 1, 5, 10, 20, 30, or 40 Hz (with a 30 s pause between trains). Compound muscle action potentials (CMAPs) were collected with a Digidata 1322A (Molecular Devices). Peak-to-peak amplitudes were analyzed in Clampfit9.2 (Molecular Devices).

## Electrophysiological recording

Electrophysiological recording of neuromuscular transmission was performed as described previously (*Li et al., 2016*; *Zhao et al., 2017*). Left hemidiaphragms with ribs and phrenic nerve distal endings were quickly dissected from anesthetized mice and pinned on Sylgard gel in oxygenated (95% $O_2$, 5% $CO_2$), 26–28°C Ringer's solution (136.8 mM NaCl, 5 mM KCl, 12 mM $NaHCO_3$, 1 mM $NaH_2PO_4$, 1 mM $MgCl_2$, 2 mM $CaCl_2$, 11 mM d-glucose, pH 7.3). Microelectrodes (20–40 MΩ with 3 M KCl) were pierced into the center of muscle fibers. Resting membrane potentials remained stable throughout the experiment at approximately −65 to −75 mV. From each hemidiaphragm, more than 5 muscle fibers were recorded for a period longer than 3 min. Data were collected with an Axopatch 200B amplifier, digitized (10 kHz low-pass filtered) with Digidata 1322A, and analyzed in Clampfit 9.2.

## Western blot

Western blot was performed as previously described (*Shen et al., 2008*), with primary antibodies: anti-Wls (rabbit, 1:500, home-made) (*Yu et al., 2010*), Synapsin 1 (rabbit, 1:1000, CST, 5297), anti-GAPDH (mouse, 1:5000, Proteintech, Cat# 60004), and anti-α-tubulin (mouse, 1:3000, Santa Cruz,

Cat# 23948). After washing, membranes were incubated with secondary antibodies HRP-conjugated goat anti-mouse or rabbit IgG (1:5000; Thermo Fisher Scientific, Cat# 31430 and 31460). Immunoreactive bands were visualized using enhanced chemiluminescence (Thermo Fisher Scientific). Autoradiographic films were scanned with an Epson 1680 scanner, and captured images were analyzed with Image J.

## Motoneuron sorting

Spinal cords were freshly isolated from HB9:Cre;tdTomato mice (P0) with cold $Ca^{2+}/Mg^{2+}$ free PBS. Tissues were minced to small pieces with scissors and digested by 0.25% Trypsin-EDTA at 37℃ for 15–30 min. Trypsin activity was stopped by adding the same amount of 10% FBS/DMEM. Cells were dissociated from each other by repeated pipetting. After filtering through 40 µm mesh (BD, Becton, Dickinson and Company), cells were spun down by centrifugation at 1000 rpm for 3 min at 4℃ and suspended in the sorting buffer ($Ca^{2+}/Mg^{2+}$ free PBS with 1 mM EDTA; 25 mM HEPES, pH 7.0, 1% FBS). GFP-labeled cells were isolated by fluorescent-activated cell sorting (FACS).

## Real-time PCR

RNA was isolated with Trizol as previously described (*Tao et al., 2013*). Two micrograms of RNA were reverse-transcribed with iScript cDNA synthesis kit (Bio-Rad). Real-time PCR was performed with SYBR Green qPCR Master Mix (Thermo Fisher Scientific). PCR primers of Wnt ligands were previously described (*Zhang et al., 2012*). RNA levels were normalized with internal controls (GADPH) that were assayed simultaneously on same reaction plates.

## Injection of Wnt proteins

Indicated Wnt proteins (10 ng/µl, 20 µl) (BD Company) were injected into right tibialis anterior (TA) with Hamilton syringe, once every three days. As control, TA muscles of left hindlimb were injected with vehicle. Thirty days later, mice were subjected for CMAP analysis, or TA muscles were collected for NMJ image analysis.

## Statistics

Data were analyzed by two-tailed unpaired Student's t test and one or two-way ANOVA. Unless otherwise indicated, data were expressed as mean ±SEM. p value less than 0.05 were considered significant.

## Acknowledgments

We are grateful to histology core in the Augusta University and Zhejiang University for EM analysis; the members from Mei-Xiong and Shen laboratories for suggestions. This work was supported in part by grants from the National Key Research and Development Program of China (2017YFA0104903 to SC); the Zhejiang Provincial Natural Science Foundation of China (LR17H090001 to SC); the National Natural Science Foundation of China (31671040 to SC, 31701036 to KZ); and the National Institutes of Health (DE15654 and DE26936 to WH; NS082007, NS090083 and AG051510 to LM; and AG045781 to WCX).

## Additional information

### Funding

| Funder | Grant reference number | Author |
| --- | --- | --- |
| National Key Research and Development Program of China | 2017YFA0104903 | Chengyong Shen |
| Natural Science Foundation of Zhejiang Province | LR17H090001 | Chengyong Shen |
| National Natural Science Foundation of China | 31671040 | Chengyong Shen |

| National Natural Science Foundation of China | 31701036 | Kejing Zhang |
| National Institutes of Health | DE15654 and DE26936 | Wei Hsu |
| National Institutes of Health | AG045781 | Wen-Cheng Xiong |
| National Institutes of Health | NS082007, NS090083, AG051510 | Lin Mei |

The funders had no role in study design, data collection and interpretation, or the decision to submit the work for publication.

## Author contributions

Chengyong Shen, Conceptualization, Formal analysis, Funding acquisition, Investigation, Writing—original draft, Writing—review and editing; Lei Li, Conceptualization, Formal analysis, Writing—original draft, Writing—review and editing; Kai Zhao, Conceptualization, Formal analysis, Investigation, Writing—original draft, Writing—review and editing; Lei Bai, Conceptualization, Formal analysis, Investigation, Writing—review and editing; Ailian Wang, Kejing Zhang, Formal analysis, Investigation, Writing—review and editing; Xiaoqiu Shu, Yatao Xiao, Jianmin Zhang, Investigation, Writing—review and editing; Tiankun Hui, Wenbing Chen, Formal analysis; Bin Zhang, Wei Hsu, Resources, Writing—review and editing; Wen-Cheng Xiong, Conceptualization, Supervision, Funding acquisition, Writing—review and editing; Lin Mei, Conceptualization, Supervision, Funding acquisition, Writing—original draft, Writing—review and editing

## Author ORCIDs
Chengyong Shen http://orcid.org/0000-0001-8184-8190
Wen-Cheng Xiong http://orcid.org/0000-0001-9071-7598
Lin Mei http://orcid.org/0000-0001-5772-1229

## Ethics

Animal experimentation: Experiments with animals were approved by Institutional Animal Care and Use Committees of Augusta University (2011-0393), Case Western Reserve University (2017-0115), and Zhejiang University (10262).

## Decision letter and Author response
Decision letter https://doi.org/10.7554/eLife.34625.017
Author response https://doi.org/10.7554/eLife.34625.018

## Additional files

### Supplementary files
• Transparent reporting form
DOI: https://doi.org/10.7554/eLife.34625.015

### Data availability
All data generated or analysed during this study are included in the manuscript and supporting files.

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
