## [Decision Letter]

[Editors’ note: this article was originally rejected after discussions between the reviewers, but the authors were invited to resubmit after an appeal against the decision.]

Thank you for submitting your manuscript "A novel autonomous regulation by Wnt for presynaptic development in the neuromuscular junction to *eLife*. Three experts carefully reviewed the manuscript. While the reviewers found the work interesting, the number of substantive questions raised was such that we feel we must reject it. We hope that the reviewers' comments will be useful to you. We apologize for not being able to deliver better news, and we hope that you will continue to consider *eLife* for future submissions.

Reviewer #1:

Synapse formation between neurons and their follower cells is an example of extreme subcellular differentiation at the site of their contact. It is driven by reciprocal interactions of pre- and postsynaptic cells. However, the underlying molecular and cellular mechanisms are poorly understood, even at the best investigated synapse, the neuromuscular junction (NMJ). The crucial factor from the motor neuron driving postsynaptic muscle differentiation is neural agrin acting through its receptor/effector LRP4 and RTK MuSK. In addition to driving postsynaptic muscle membranes in response to MuSK activation muscle fibers are thought to act back on motor neurites to drive nerve terminal differentiation through mechanisms that may involve Wnt signalling.

In recent years, evidence, some of it conflicting, for an involvement of Wnts, in various ways, in modulating rodent NMJ formation has been accumulating. Investigating their involvement in NMJ formation has been complicated by the fact, that some 18 Wnts have been identified in mammals. In the present paper, authors demonstrate by genetic approaches that Wnts secreted from motor neurons is involved in presynaptic terminal differentiation and function. Specifically, genetic deletion of Wnt ligand secretion mediator, Wls, in motor neurons, but not in Schwann cells or in muscle fibers, impair functional differentiation and maintenance of rodent NMJs. As Wls is required for sorting and transport to the cell surface and, thus, for secretion of various Wnts, these data indicate that Wnts secreted from motor neurons are involved in NMJ maintenance. Specifically, evidence is presented that the Wnt required for NMJ maintenance is Wnt 7A. Reinnervation experiments further show that Wls mutant motor axons show a deficit in their capacity to reinnervate old endplate sites upon severing the nerve. This paper is thus an important contribution towards full understanding of NMJ formation and implicates dysregulation of Wnt signalling as a potential mechanism of certain neuromuscular disorders.

Although overall, the data support the conclusion, I feel that some of the mechanisms proposed appear overinterpreted. Attention should be given to phrasing that accurately reflects what the figures actually show. The points below can probably be dealt with readily.

Figure 1D: Authors conclude from grip strength reduction in Wls mutants that muscle function is impaired. For a meaningful comparison of muscle function grip strengths should be compared not only between adults of same age, but also between animals of same weight! Part of the grip strength reduction in mutants could arise from lower muscle mass associated with lower body weight of the latter.

Results section: NMJ formation appears………: It is stated that NMJs are characterized in mouse embryos. Figure 2 suggests that images were taken from older animals: fetuses or newborn?

The resolution of Figure 2A is not sufficient to show that morphologies of nerve terminals were comparable. – Note: typo: "defasciculation".

Age-dependent……: "The mild phenotypes…….". Are the phenotypes mild or absent? P- Values in the para above suggests they NMJ parameters are indistinguishable. Either improve statistics by increasing sample size in Figure 2C or accept that phenotypes are indistinguishable.

In Figure legend 2, the age of the animals should be given.

Figure 3: In panel B, calibration bar is missing. Varicosities similar to those seen primarily in Wls mutants are also observed in NMJs recovering from Botulinus toxin poisoning. Thus, they may be indicative of reinnervation in response to functional denervation, as is suggested by neuromuscular transmission failure in the mutants (Figure 5). This should be mentioned in the Discussion section.

It is striking that at two months, SV2 staining in w.t. muscle is still strong, as in Figure 3A. Lupa and Hall, (1989) have described that unlike during fetal development, SV2 is no longer visible in motor axons of adult animals (>2 weeks postnatal). Have the authors made out a difference between SV2 stainings in w.t. vs Wls mutant axons, as is suggested by the Western blot in Figure 6D? This could be important for the interpretation of varicosity formation potentially reflecting axon regeneration in Wls mutants. Discuss!

In Figure legend 3, the age of the animals should be given.

Reduced synaptic vesicles……..: It is not clear, how from EM sections such as those in Figure 4A centre, with their inhomogenous distribution of synaptic vesicles in the nerve terminals, synaptic vesicle densities can be derived quantitatively (as claimed in Figure 4B). Describe!. Were there also mutant terminals with normal vesicle densities, as would be expected from high SV2 aggregates opposed to high density AChR clusters (see Figure 6B)? Furthermore, is the lack of a basal lamina in the synaptic cleft (as shown in Figure 4A bottom), commonly observed in the mutants? It could be related to loss of postsynaptic structures.

Subsection “Impaired neuromuscular transmission”: it is stated that "postsynaptic deficits were likely caused by reduced neuromuscular transmission". This interpretation is unlikely to be correct. For example, in AChR deficiency myasthenic syndrome (through a mutation in the AChRe promoter (specifically its N-box) with severely impaired transmission, postsynaptic membrane appears normal.

Clustered synaptic vesicles: based on differences in SV2 stainings, in their localized distribution in nerve terminals and accumulations in varicosities, and the higher SV2 immunoreactivity in sciatic nerves (Figure 6B, C, D) between w.t. and Wls mutant mice, it is suggested that synaptic vesicle transport may be impaired in Wls ablated axons, which would explain the lower vesicle density in the mutant nerve terminals. Although SV2 is considered a vesicle-specific protein, I find it risky to conclude from local aggregations in SV2 staining intensity the local aggregation of synaptic vesicles. For confirmation, EM analysis of vesicle densities in axonal varicosities would be required. Furthermore, in Figure 6B, some of the strong presynaptic SV2 accumulations in Wls mutants are opposed to strong AChR clusters, which is not consistent with the EM data in Figure 4. Explain!

Statistical tests: For every experiment analysed by statistical tests, authors should give sample sizes: e.g. how many mepp amplitudes were sampled for calculation of a mean {plus minus} S.E. When were paired or unpaired t-tests used?

Reviewer #2:

This manuscript reported results to support the conclusion that Wnt signaling pathways act autonomously in presynaptic development at the neuromuscular junction (NMJ). The key approach is genetic deletion of the Wnt ligand secretion mediator (Wls) gene in motor neurons, muscles or Schwann cells. The analysis and the results in the current version of manuscript have not convincingly supported the conclusion.

1) Even though no detectable NMJ defects were found in the HSA::Cre and Wnt1::Cre Wls mutants, it would be crucial to show Wls is indeed completely absent in muscles and Schwann cells and there is no possibility that other Wls-like molecules can compensate for the loss. In particular, muscle fibers contain multiple nuclei and the reporter system (td-Tomato) cannot prove a complete deletion of Wls in all nuclei because the reporter molecule will diffuse and label the entire muscle fiber. It is a lingering concern for the HAS::Cre line when used in studying the NMJ at early developmental stages.

2) As Wnts are presumably secreted and bind their cognate receptors to function, a central question is what Wnts are expressed and secreted from muscles and Schwann cells. For example, does either cell type express Wnt 7A?

3) The HB9::Cre Wls mutants display grip strength deficits. But the analysis of the NMJ has only been done in the diaphragm muscle. It would be important to analyze other muscles that are directly relevant to grip strength phenotype.

4) If exogenous ACh agonists are applied, is the endplate potential normal?

5) The defect in synaptic vesicle aggregation is interesting. The authors tried to show increased SV2 protein levels in the sciatic nerve to support the finding, but the Figure 6C is not convincing. Since the sciatic nerve is mixed with many non-motor axons, it would have been more convincing to use pure motor nerves, e.g. phrenic nerve.

6) The nerve transplantation experiment is interesting, but it is less relevant to the conclusion. They can be completely removed.

7) The partial rescue of NMJ defects via injecting individual Wnt proteins is interesting but the results do not directly support the conclusion on cell autonomous Wnt signaling. It is a little bit hard to interpret the results without knowing what relevant Wnt receptors are expressed in cells involved in NMJ formation or maintenance. For example, even Wnt7A expression/secretion from motor neurons is reduced, it cannot rule out that motor neuron-derived Wnt7A activates muscle (or Schwann cell) pathways to elicit retrograde signals to promote presynaptic development unless the receptor(s) for Wnt7A (or other relevant Wnt molecules) is selectively deleted from motor neurons, not other cell types.

Reviewer #3:

Here, Shen et al. report a very interesting study on a novel role for Wnt in presynaptic assembly in the neuromuscular junction. The authors used the main Wnt secretion mediator Wls to specifically abolish Wnt secretion in each of the three NMJ compartments (skeletal muscle, motor neurons and Schwann cells). They showed that only Wls deletion in motor neurons leads to age-dependent NMJ deficits and neurotransmission impairment in a cell-autonomous manner. Using a combination of morphological analysis, electron microscopy and electrophysiology, they characterized the NMJ deficits and nicely demonstrate that motoneuronal specific deletion of Wls results in progressive NMJ degeneration, muscle weakness, impaired neurotransmission, motor axonal swelling and synaptic vesicle reduction. They also showed that motoneuronal Wls is required for NMJ regeneration. From there, they screened for the Wnt ligands that are expressed in motoneurons and identified those whose secretion is regulated by Wls in heterologous HEK cells. Wnt5A and 7A were the most sensitive to Wls mutation. Injection of recombinant Wnt7A, but not Wnt5A in TA muscle can partially rescue the morphological presynaptic deficits of HB9-Wl-/-. Overall this is a very interesting study, the analysis of data was carefully performed and most of the main conclusions are supported by the provided results. However, although the finding that Wnt7A may play a role in presynaptic assembly is novel, the current study does not provide sufficient evidence to support this conclusion and it remains unclear how Wnts secretion from motoneurons contributes to presynaptic functions resulting in age-dependent NMJ dysfunction when altered.

The authors state that Wls deletion in motoneurons, but not skeletal muscle or Schwann cells shows NMJ defects. Given that several Wnts identified by the authors in motoneurons are known to regulate postsynaptic differentiation, some of them being low or not sensitive to Wls, there is a possibility that Wnts secreted by the motoneurons may have compensated the muscle and/or Schwann cells NMJ phenotypes. A more detailed analysis of the NMJ phenotypes during NMJ formation, including combined HSA and/or HB9 and/or Wnt1-Wls-/- should be performed to support their conclusions. I realized that it is a demanding task, but this is not unfeasible.

The authors identified several Wnts molecules expressed in motoneurons and showed that apart from Wnt7A and Wnt5A, several other (Wnt7B and Wnt9B) were sensitive to Wls. It is not clear why these Wnts were not tested in the rescue experiment. Also, how Wnt overexpression in the muscle can rescue the presynaptic NMJ deficits? Neurotransmission analysis could be performed following Wnt7A injection.

The authors previously reported that Wnt7A modulates agrin-induced AChR clustering in muscle cell (Barik et al., 2014) suggesting that Wnt7A regulates postsynaptic differentiation. This should be included and commented in the manuscript.

Developmental, post-natal stages and muscle types used should be mentioned in the figure legends.

In several figures the title is misleading. The authors should replace Wl-/- by HB9-Wl-/-.

Figure 5 and Figure 3—figure supplement 1Figure: Please clarify the title.

Figure 3—figure supplement 1Figure: Please clarify how synapse elimination was evaluated.

Figure 8D: The graph legend is missing.

Figure 1C: Co-staining of Wls with a motoneuron marker should be performed to support the conclusion.

---

## [Author Response]

[Editors’ note: the author responses to the first round of peer review follow.]

Reviewer #1:Synapse formation between neurons and their follower cells is an example of extreme subcellular differentiation at the site of their contact. It is driven by reciprocal interactions of pre- and postsynaptic cells. However, the underlying molecular and cellular mechanisms are poorly understood, even at the best investigated synapse, the neuromuscular junction (NMJ). The crucial factor from the motor neuron driving postsynaptic muscle differentiation is neural agrin acting through its receptor/effector LRP4 and RTK MuSK. In addition to driving postsynaptic muscle membranes in response to MuSK activation muscle fibers are thought to act back on motor neurites to drive nerve terminal differentiation through mechanisms that may involve Wnt signalling.In recent years, evidence, some of it conflicting, for an involvement of Wnts, in various ways, in modulating rodent NMJ formation has been accumulating. Investigating their involvement in NMJ formation has been complicated by the fact, that some 18 Wnts have been identified in mammals. In the present paper, authors demonstrate by genetic approaches that Wnts secreted from motor neurons is involved in presynaptic terminal differentiation and function. Specifically, genetic deletion of Wnt ligand secretion mediator, Wls, in motor neurons, but not in Schwann cells or in muscle fibers, impair functional differentiation and maintenance of rodent NMJs. As Wls is required for sorting and transport to the cell surface and, thus, for secretion of various Wnts, these data indicate that Wnts secreted from motor neurons are involved in NMJ maintenance. Specifically, evidence is presented that the Wnt required for NMJ maintenance is Wnt 7A. Reinnervation experiments further show that Wls mutant motor axons show a deficit in their capacity to reinnervate old endplate sites upon severing the nerve. This paper is thus an important contribution towards full understanding of NMJ formation and implicates dysregulation of Wnt signalling as a potential mechanism of certain neuromuscular disorders.Although overall, the data support the conclusion, I feel that some of the mechanisms proposed appear overinterpreted. Attention should be given to phrasing that accurately reflects what the figures actually show. The points below can probably be dealt with readily.

We thank reviewer 1 for the remarks that “This paper is thus an important contribution towards full understanding of NMJ formation”; “overall, the data support the conclusion” and his/her constructive critiques and suggestions that have significantly improved the manuscript.

Figure 1D: Authors conclude from grip strength reduction in Wls mutants that muscle function is impaired. For a meaningful comparison of muscle function grip strengths should be compared not only between adults of same age, but also between animals of same weight! Part of the grip strength reduction in mutants could arise from lower muscle mass associated with lower body weight of the latter.

This is a good suggestion. The revision also compared grip strength between two groups of mice with similar body weights (29.3 + 1.1 g vs 27.7 + 1.6 g for control and mutants, respectively, p > 0.05). As shown in revised Figure 2G, HB9-Wls^-/-^ mice had reduced grip strength (129 + 9.7 g in control mice; 71.8 + 7.9 g in mutant mice; p < 0.01) (–subsection “Reduced body weight and muscle strength in Wls motoneuron knockout mice”).

Results section: NMJ formation appears………: It is stated that NMJs are characterized in mouse embryos. Figure 2 suggests that images were taken from older animals: fetuses or newborn?

Sorry for the confusion. It shows NMJs of E18.5 mice. More representative, high-resolution images are provided in revised Figure 2—figure supplement 2A.

The resolution of Figure 2A is not sufficient to show that morphologies of nerve terminals were comparable. – Note: typo: "defasciculation".

Corrected.

Age-dependent……: "The mild phenotypes…….". Are the phenotypes mild or absent? P- Values in the para above suggests they NMJ parameters are indistinguishable. Either improve statistics by increasing sample size in Figure 2C or accept that phenotypes are indistinguishable.

Good point. The reviewer is correct that there were no detectable deficits of NMJs in mutant embryonic and P0 mice (in terms of gross morphology, endplate width, AChR size, and NMJ density). “mild” was a word of bad choice. This sentence has now been revised as “Normal NMJs at P0 pups were unable to explain…” (subsection “Age-dependent nerve terminal swellings and poor innervation”).

In Figure legend 2, the age of the animals should be given.

Good point. Information on ages of animals has now provided to legend to revised Figure 2—figure supplement 2 and all other figures in revised manuscript.

Figure 3: In panel B, calibration bar is missing.

Calibration bar was added in revised Figure 3B.

Varicosities similar to those seen primarily in Wls mutants are also observed in NMJs recovering from Botulinus toxin poisoning. Thus, they may be indicative of reinnervation in response to functional denervation, as is suggested by neuromuscular transmission failure in the mutants (Figure 5). This should be mentioned in the Discussion section.

Good suggestion. This is now discussed as suggested (Discussion section).

It is striking that at two months, SV2 staining in w.t. muscle is still strong, as in Figure 3A. Lupa and Hall, (1989) have described that unlike during fetal development, SV2 is no longer visible in motor axons of adult animals (>2 weeks postnatal). Have the authors made out a difference between SV2 stainings in w.t. vs Wls mutant axons, as is suggested by the Western blot in Figure 6D? This could be important for the interpretation of varicosity formation potentially reflecting axon regeneration in Wls mutants. Discuss!

The reviewer is correct that SV2 staining is weak in axons at ages of 2 weeks and old although remains positive at terminals. This is in agreement with our data in Figure 6B where SV2 signal was weak in axons at two months of mice (note axons could be readily labeled by anti-neurofilament antibody). Please note that NMJ images in Figure 3A were obtained with antibodies also against neurofilament and thus showed strong axon staining.

We would like to point out that SV2 staining was detectable in axons of wild type mice at ages of two months, but the staining was sparse (Figure 6B, empty arrowhead). However, SV2 was increased in axons of adult mutant mice by staining and Western blot analysis (Figure 6B and Figure 6C). This is similar to a previous report that enhancement of synaptic vesicle proteins in sciatic nerves when autophagy is impaired (Luningschror et al., 2017).

In addition to increased SV2 levels, SV2-positive aggregates were detected in Wls mutants (Figure 6B). Interestingly, such varicosities were also observed in NMJs recovering from Botulinum toxin poisoning (Rogozhin et al., 2008). Together, these observations are indicative of reinnervation in response to functional denervation. As suggested, the revised manuscript added a brand-new paragraph to discuss implication of these results (Discussion section). We thank the reviewer for this suggestion.

In Figure legend 3, the age of the animals should be given.

Good point. Information on ages of animals has now provided to legend to Figure 3 and all other figures in revised manuscript.

Reduced synaptic vesicles……..: It is not clear, how from EM sections such as those in Figure 4A centre, with their inhomogenous distribution of synaptic vesicles in the nerve terminals, synaptic vesicle densities can be derived quantitatively (as claimed in Figure 4B). Describe!.

Sorry for the oversight. Synaptic vesicles in randomly-selected areas within 500 nm from presynaptic membrane were quantified. Junctional folds and synaptic cleft width were quantified in randomly-selected areas where more than 5 consecutive junctional folds could be visibly defined. This information is now provided in the revised manuscript (subsection “Electron microscopy (EM) analysis”).

Were there also mutant terminals with normal vesicle densities, as would be expected from high SV2 aggregates opposed to high density AChR clusters (see Figure 6B)?

Yes, some nerve terminals in mutant mice appeared to be normal (~47%) (Figure 3A, 3E and 3F).

Furthermore, is the lack of a basal lamina in the synaptic cleft (as shown in Figure 4A bottom), commonly observed in the mutants? It could be related to loss of postsynaptic structures.

Good point. As shown in Figure 4A, in control NMJs, basal lamina was visible in synaptic cleft and between junctional folds, with highest electron density in the middle (empty arrowheads). However, basal lamina density was reduced and poorly defined in mutant mice. This could be a mechanism of loss of postsynaptic structures. The revised manuscript now describes findings on basal lamina in Results section and discussed their implications in Discussion section.

Subsection “Impaired neuromuscular transmission”: it is stated that "postsynaptic deficits were likely caused by reduced neuromuscular transmission". This interpretation is unlikely to be correct. For example, in AChR deficiency myasthenic syndrome (through a mutation in the AChRe promoter (specifically its N-box) with severely impaired transmission, postsynaptic membrane appears normal.

We agree. The revised Discussion section has discussed alternative interpretations. For example, it may be caused by reduced and poorly organized synaptic basal lamina.

Clustered synaptic vesicles: based on differences in SV2 stainings, in their localized distribution in nerve terminals and accumulations in varicosities, and the higher SV2 immunoreactivity in sciatic nerves (Figure 6B, C, D) between w.t. and Wls mutant mice, it is suggested that synaptic vesicle transport may be impaired in Wls ablated axons, which would explain the lower vesicle density in the mutant nerve terminals. Although SV2 is considered a vesicle-specific protein, I find it risky to conclude from local aggregations in SV2 staining intensity the local aggregation of synaptic vesicles. For confirmation, EM analysis of vesicle densities in axonal varicosities would be required. Furthermore, in Figure 6B, some of the strong presynaptic SV2 accumulations in Wls mutants are opposed to strong AChR clusters, which is not consistent with the EM data in Figure 4. Explain!

We agree with the reviewer’s comments. First, we have performed additional EM analysis; however, we have encountered a technical difficulty. NMJ occupies 0.01-0.1% of muscle fiber surface. Presynaptic vesicles/membrane are identified by their proximity to characteristic postsynaptic junctional folds. In Wls mutant mice, NMJs were disorganized and junctional folds were reduced. In addition, as shown in Figure 6B, the area of SV2+ aggregates was about 5 µm and less than 2% of a NMJ. We tried but were unable to convincingly identify vesicle clusters in axons by EM. Second, the nature of SV2+ aggregates is unknown. They may be inclusion bodies containing SV2 proteins for degradation. Third, we wish to point out that SV2 staining was low in wild type axons in adult mice, but it was increased in axons of mutant mice (Figure 6B). To test this hypothesis, we performed additional Western blot analysis of Synapsin 1, another marker of synaptic vesicles. As shown in revised Figure 6C, Synapsin 1 level was increased in Wls mutant axons, compared with control axons. This is in agreement with increased levels of SV2 in axons (Figure 6C). Together, the results suggest possible accumulation of vesicle-specific proteins in mutant axons. Although it is difficult to attribute these proteins to aggregated vesicles (because technical difficulty described above), their increased levels in axons may lead to a reduction at presynaptic terminals.

Regarding “strong presynaptic SV2 accumulations in Wls mutants”, we would like to point out that SV2 signaling is discontinuous and spotty, faint or missing in many areas where AChR was present. As described above, they may result from inclusion bodies containing SV2 proteins for degradation.

We have discussed these points in revised Discussion section.

Statistical tests: For every experiment analysed by statistical tests, authors should give sample sizes: e.g. how many mepp amplitudes were sampled for calculation of a mean {plus minus} S.E. When were paired or unpaired t-tests used?

Statistic information has now been provided in legends to all figures in revised manuscript.

Reviewer #2:This manuscript reported results to support the conclusion that Wnt signaling pathways act autonomously in presynaptic development at the neuromuscular junction (NMJ). The key approach is genetic deletion of the Wnt ligand secretion mediator (Wls) gene in motor neurons, muscles or Schwann cells. The analysis and the results in the current version of manuscript have not convincingly supported the conclusion.

We thank for reviewer 2 for the above remark that “manuscript reported results to support the conclusion that Wnt signaling pathways act autonomously in presynaptic development at the neuromuscular junction (NMJ)” and the following remarks in individual comments “The defect in synaptic vesicle aggregation is interesting…”; “The nerve transplantation experiment is interesting…” and “The partial rescue of NMJ defects via injecting individual Wnt proteins is interesting…” and his or her constructive critiques and suggestions that have significantly improved the manuscript.

1) Even though no detectable NMJ defects were found in the HSA::Cre and Wnt1::Cre Wls mutants, it would be crucial to show Wls is indeed completely absent in muscles and Schwann cells and there is no possibility that other Wls-like molecules can compensate for the loss. In particular, muscle fibers contain multiple nuclei and the reporter system (td-Tomato) cannot prove a complete deletion of Wls in all nuclei because the reporter molecule will diffuse and label the entire muscle fiber. It is a lingering concern for the HAS::Cre line when used in studying the NMJ at early developmental stages.

Good suggestions. We have performed additional experiments. As shown in revised suppl Figure 1B and 1F, levels of Wls were reduced in muscles of HSA-Wls^-/-^ mice and sciatic nerves of Wnt1-Wls^-/-^ mice, respectively, compared with control mice. The reduction in mutant muscles and sciatic nerves was not 100% because these tissues contain cells where Cre was not expressed (such as blood vessels, nerve terminals, and Schwann cells in suppl Figure 1B and axons in suppl Figure 1F). Although there was no observable defects of NMJ formation in Wnt1-Wls^-/-^ mice, we found that Wnt1-Wls^-/-^ mice died soon after birth with severe craniofacial skeleton defects, in accord with the notion that Wnt signaling in neural crests is critical for craniofacial skeleton development (Fu et al., 2011).

It is a good question whether Wls was completely knocked out by HSA::Cre. Cre in HSA::Cre mice is expressed in myotomal somites as early as E9.5, a time prior to NMJ formation (around E13.5). This line has been widely used by various laboratories to study NMJ development (Li et al., 2008; Escher et al., 2005; Jaworski et al., 2006; Wu et al., 2012; Seaberg et al. 2015). In newly provided data, Wls in HAS-Wls^-/-^ muscles was reduced to ~5% of that of control mice (revised Figure 2—figure supplement 1B). As described above, the residual Wls protein could be from other cell types in muscle tissues such as blood vessel cells and even axon terminals. In agreement with these considerations, a recent study found no apparent NMJ deficits when Wls was knocked out by Pax3::Cre (Remedio et al. 2016).

Whether there is a Wls-like molecule is an outstanding question. However, no such a molecule has been identified so far. Wls knockout is embryonic lethal (Fu et al. 2009). Its conditional knockout has been shown to cause deficits in various tissues including gut (Valenta et al., 2016), spinal cord (Onishi et al. 2018), and craniofacial and brain development (Fu et al., 2011). This paper focuses on whether and which Wnts from motoneurons contribute to NMJ development. We would hope that the reviewer would agree whether there is Wls-like molecules could be addressed in a future study. This point has been discussed in the revised manuscript (Discussion section).

2) As Wnts are presumably secreted and bind their cognate receptors to function, a central question is what Wnts are expressed and secreted from muscles and Schwann cells. For example, does either cell type express Wnt 7A?

We performed additional experiments to analyze expression pattern of all 19 Wnts in Schwann cells and muscle cells. Dominant Wnts appeared to be Wnt5A and Wnt9A in Schwann cells and Wnt4, Wnt6 and Wnt9A in muscle cells (revised Figure 1). Wnt7A is not highly expressed in muscle cells [in agreement with an earlier report (Zhang et al., 2012)] or Schwann cells.

3) The HB9::Cre Wls mutants display grip strength deficits. But the analysis of the NMJ has only been done in the diaphragm muscle. It would be important to analyze other muscles that are directly relevant to grip strength phenotype.

Sorry for not being clear. In addition to diaphragm, NMJ analysis has been done in gastrocnemius (revised Figure 3A-3E, Figure 5A-5C, Figure 6A and 6B, and Figure 3—figure supplement 1) and tibialis anterior (Figure 7C-7E). These muscles are now clearly indicated in respective figure legends.

4) If exogenous ACh agonists are applied, is the endplate potential normal?

Good question. In newly performed experiments, we treated diaphragms with CCh (10 µM), an ACh agonist, for 2 min and observed its effects on mEPPs. Consistent with previous reports (Katz et al., 1972; Miyamoto et al. 1974), CCh could enhance mEPP frequency and amplitude in control diaphragms (revised Figure 5D-5I). Interestingly, CCh also elevated mEPP frequency and amplitude in Wls mutant diaphragms to similar or higher levels of controls in the absence of CCh (Figure 5D-5I). Nevertheless, mEPP frequency and amplitude in the presence of CCh were lower in mutant, compared with control at age of P30. At P1, mEPP amplitudes in the absence of CCh were similar between HB9-Wls^-/-^ and control muscles (Figure 5H). These results indicate that transmission impairment in mutant mice could be attenuated by ACh agonist.

5) The defect in synaptic vesicle aggregation is interesting. The authors tried to show increased SV2 protein levels in the sciatic nerve to support the finding, but the Figure 6C is not convincing. Since the sciatic nerve is mixed with many non-motor axons, it would have been more convincing to use pure motor nerves, e.g. phrenic nerve.

Good suggestion. We now performed additional Western blot with phrenic nerves for SV2 and included another vesicle marker Synapsin 1. As shown in revised Figure 6C, both SV2 and Synapsin 1 were increased in phrenic nerves of mutant mice.

6) The nerve transplantation experiment is interesting, but it is less relevant to the conclusion. They can be completely removed.

We agree that this result is less relevant and has been completely removed, as suggested.

7) The partial rescue of NMJ defects via injecting individual Wnt proteins is interesting but the results do not directly support the conclusion on cell autonomous Wnt signaling. It is a little bit hard to interpret the results without knowing what relevant Wnt receptors are expressed in cells involved in NMJ formation or maintenance. For example, even Wnt7A expression/secretion from motor neurons is reduced, it cannot rule out that motor neuron-derived Wnt7A activates muscle (or Schwann cell) pathways to elicit retrograde signals to promote presynaptic development unless the receptor(s) for Wnt7A (or other relevant Wnt molecules) is selectively deleted from motor neurons, not other cell types.

How motoneuron Wnts regulate NMJ formation is a great question. They could act via activating Wnt receptors on motoneurons, on muscles and on Schwann cells. To address this question, we performed additional experiments to determine the expression patterns of Wnt receptors including 10 Frizzled proteins. As shown in revised Figure 1—figure supplement 1, many were detectable in the ventral horns of the spinal cord, Schwann cells, and muscle cells. Figuring out which Frizzled (plus Wnt co-receptors LPR5/LPR6) is involved in what cells could be a daunting task.

This paper provides first genetic evidence that Wnts from motoneurons are critical for NMJ development. Effects could be mediated by Wnt receptors on motoneurons, Schwann cells and/or muscle fibers, as the reviewer pointed out. Considering that our paper has 7 figures and 6 supplemental figures, each with multiple panels, we would hope that the reviewer would agree that how motoneuron Wnts act could be addressed in a future study.

We deleted Wls in the motoneuron and found the defects in the motoneuron, which we believe it is the cell-autonomous effect of Wls. But we agree that we don’t know whether it is through a cell-autonomous molecular mechanism without knowing the receptors. To get rid of the confusion, we have avoided using the term – cell autonomous – in the revised manuscript and also changed the paper title. We thank the reviewer for this question.

Reviewer #3:Here, Shen et al. report a very interesting study on a novel role for Wnt in presynaptic assembly in the neuromuscular junction. The authors used the main Wnt secretion mediator Wls to specifically abolish Wnt secretion in each of the three NMJ compartments (skeletal muscle, motor neurons and Schwann cells). They showed that only Wls deletion in motor neurons leads to age-dependent NMJ deficits and neurotransmission impairment in a cell-autonomous manner. Using a combination of morphological analysis, electron microscopy and electrophysiology, they characterized the NMJ deficits and nicely demonstrate that motoneuronal specific deletion of Wls results in progressive NMJ degeneration, muscle weakness, impaired neurotransmission, motor axonal swelling and synaptic vesicle reduction. They also showed that motoneuronal Wls is required for NMJ regeneration. From there, they screened for the Wnt ligands that are expressed in motoneurons and identified those whose secretion is regulated by Wls in heterologous HEK cells. Wnt5A and 7A were the most sensitive to Wls mutation. Injection of recombinant Wnt7A, but not Wnt5A in TA muscle can partially rescue the morphological presynaptic deficits of HB9-Wl-/-. Overall this is a very interesting study, the analysis of data was carefully performed and most of the main conclusions are supported by the provided results. However, although the finding that Wnt7A may play a role in presynaptic assembly is novel, the current study does not provide sufficient evidence to support this conclusion and it remains unclear how Wnts secretion from motoneurons contributes to presynaptic functions resulting in age-dependent NMJ dysfunction when altered.

We thank the reviewer for the remarks that “Shen et al., report a very interesting study on a novel role for Wnt in presynaptic assembly in the neuromuscular junction” and that “Overall this is a very interesting study, the analysis of data was carefully performed and most of the main conclusions are supported by the provided results” and his or her constructive critiques and suggestions that have significantly improved the manuscript.

The authors state that Wls deletion in motoneurons, but not skeletal muscle or Schwann cells shows NMJ defects. Given that several Wnts identified by the authors in motoneurons are known to regulate postsynaptic differentiation, some of them being low or not sensitive to Wls, there is a possibility that Wnts secreted by the motoneurons may have compensated the muscle and/or Schwann cells NMJ phenotypes. A more detailed analysis of the NMJ phenotypes during NMJ formation, including combined HSA and/or HB9 and/or Wnt1-Wls-/- should be performed to support their conclusions. I realized that it is a demanding task, but this is not unfeasible.

The reviewer wanted us to determine whether motoneuron Wnts could compensate Wls deletion in muscles or Schwann cells. To test whether motoneuron Wnts could compensate Wls deletion in muscles, we generated mice where Wls was knocked out in both motoneurons and muscles (HB9/HSA-Wls^-/-^). As shown in revised Figure 2—figure supplement 3A and 3B, NMJ morphology in these mice was similar to that of HSA-Wls^-/-^ mice, suggesting that motoneuron Wnts had little compensating effects on Wls deletion in muscles.

Along the same line, to determine whether motoneuron Wnts compensate Wls deletion in Schwann cells, we characterized HB9/Wnt1-Wls^-/-^ mice where Wls was deleted in both motoneurons and Schwann cells. As shown in revised Figure 2—figure supplement 3C and 3D, NMJ morphology in these mice was also similar to that of Wnt1-Wls^-/-^ mice, suggesting that motoneuron Wls could not compensate Wls deletion in Schwann cell.

We have tried hard to obtain triple-cell knockout mice (i.e., Wls deletion in motoneurons, muscles and Schwann cells). However, because Wnt1-Wls^-/-^ mice die at birth, it has been very difficult to generate sufficient numbers of HB9/HSA/Wnt1-Wls^-/-^ mice.

In newly performed experiments, we checked expression of 19 Wnts in motoneurons, Schwann cells, and muscle cells. Noticeably, Wnt7A appeared to be the most abundant, followed by Wnt3, Wnt4, and Wnt7B in motoneurons. Expression of other Wnts appeared to be low in motoneurons. Dominant Wnts in Schwann cells appeared to be Wnt5A and Wnt9A whereas Wnt4, Wnt6 and Wnt9A were most abundant in muscle cells (revised Figure 1). By knocking out Wls in motoneurons, we provide first genetic evidence that Wnts from motoneurons are critical for NMJ. In revision, we avoided implications that Wnts from muscles or Schwann cells are not important. The revision has discussed these points in revised Discussion section.

The authors identified several Wnts molecules expressed in motoneurons and showed that apart from Wnt7A and Wnt5A, several other (Wnt7B and Wnt9B) were sensitive to Wls. It is not clear why these Wnts were not tested in the rescue experiment. Also, how Wnt overexpression in the muscle can rescue the presynaptic NMJ deficits? Neurotransmission analysis could be performed following Wnt7A injection. The authors previously reported that Wnt7A modulates agrin-induced AChR clustering in muscle cell (Barik et al., 2014) suggesting that Wnt7A regulates postsynaptic differentiation. This should be included and commented in the manuscript.

These are good questions/suggestions.

1) We performed new experiments and demonstrated that Wnt7B, like Wnt7A, can also rescue; but Wnt9B, like Wnt5A, cannot rescue the defects in HB9-Wls^-/-^ mice (revised Figure 7C and 7D).

2) “How Wnt overexpression in the muscle can rescue the presynaptic NMJ deficits?”

- Because Wls was mutated in motoneurons in HB9-Wls^-/-^ mice, motoneurons are unable to process Wnt proteins for secretion. Therefore, we provided Wnts in muscles.

Exactly how motoneuron Wnts regulate NMJ development is a great question. They could act via activating Wnt receptors on motoneurons, on muscles and on Schwann cells. We performed additional experiments to determine the expression patterns of Wnt receptors including 10 Frizzled proteins. As shown in revised Figure 1—figure supplement 1, many were detectable in the ventral horns of the spinal cord, Schwann cells, and muscle cells. Figuring out which Frizzled (plus Wnt co-receptors LPR5/LPR6) is involved in what cells could be a daunting task.

This paper provides first genetic evidence that Wnts from motoneurons are critical for NMJ maintenance. Without Wnts, synaptic vesicles were deposited ectopically and reduced at nerve terminals, and these dysfunctional synapses were gradually degenerated with a feature of axonal swelling. The effects could be mediated by Wnt receptors on motoneurons, Schwann cells and/or muscle fibers. Considering that the paper has packed 7 figures and 6 supplemental figures, each with multiple panels, we would hope that the reviewer would agree that how motoneuron Wnts act could be addressed in a future study. We have included this in the revised Discussion section.

3) “Neurotransmission analysis could be performed following Wnt7A injection.”

- We performed new experiments and found that impaired neurotransmission in the HB9-Wls^-/-^ was rescued by Wnt7A injection (revised Figure 7E).

4) The authors previously reported that Wnt7A modulates agrin-induced AChR clustering in muscle cell (Barik et al., 2014) suggesting that Wnt7A regulates postsynaptic differentiation. This should be included and commented in the manuscript.

- Yes, a previous in vitro study of ours showed that Wnt7A could reduce the number of agrin-induced AChR clusters in C2C12 myotubes. This result could suggest that Wnt7A may act by regulating postsynaptic differentiation. However, we did not observe a change in AChR cluster size or intensity in mice after Wnt7A injection (data not shown) although this reduced the number of axonal swellings. This suggests that Wnt7A could act directly on nerve terminals in the HB9-Wls^-/-^ mutant. The revised manuscript discussed these possibilities (Discussion section).

Developmental, post-natal stages and muscle types used should be mentioned in the figure legends.

Sorry for the oversight. Such information has now been provided in revision in figure legends.

In several figures the title is misleading. The authors should replace Wl-/- by HB9-Wl-/-. Figure 5 and Figure 3—figure supplement 1: Please clarify the title.

We have gone through figures and text to ensure HB9-Wls^-/-^ is used.

Figure 3—figure supplement 1: Please clarify how synapse elimination was evaluated.

To visualize AChR clusters on single muscle fibers, gastrocnemius was fixed in 4% PFA and stained with R-BTX and DAPI to show AChR clusters and myonuclei. Muscles were washed in PBS and teased into single fibers and mounted with Vectashield mounting medium. We counted the number of AChR cluster in each single muscle fiber and defined that synapse elimination was impaired when more than one AChR cluster was found after P15. Detailed description of synapse elimination was provided in the revision (Figure 3—figure supplement 1).

Figure 8D: The graph legend is missing.

The legend was provided in the revised Figure 7C.

Figure 1C: Co-staining of Wls with a motoneuron marker should be performed to support the conclusion.

Thanks for the suggestion. We performed new experiments to co-stain Wls with a motoneuron marker ChAT. Wls signal was diminished in the ventral horn of HB9-Wls^-/-^ mice, compared with that in control mice. The new data (revised Figure 2C) were added in the revised Results section.